# International Logistics and Cross-Border E-Commerce Trade: Who Matters Whom?

Yugang He [1] , Renhong Wu [2] and Yong-Jae Choi [1],*

1  College of Commerce, Jeonbuk National University, Jeonju 54896, Korea; 1293647581@jbnu.ac.kr
2  College of Economics, Guangdong Ocean University, Zhanjiang 524088, China; wurenhongbini@163.com
*  Correspondence: yjchoi06@jbnu.ac.kr

**Abstract:** Unlike previous papers on international logistics and cross-border e-commerce trade, this paper sets Organization for Economic Co-operation and Development (OECD) countries as an example to explore the dynamic interaction between international logistics and cross-border e-commerce trade. The panel data for the period 2000–2018 will be employed to perform an empirical analysis via a host of econometric techniques, such as panel unit root tests, panel cointegration tests, panel causality tests and the panel vector error correction model. Incorporating with other control variables, we find that there is a long-term relationship between international logistics and cross-border e-commerce trade. Specifically speaking, in the long-run, international logistics has a positive and significant effect on cross-border e-commerce trade. However, in the short-run, international logistics has a negative and significant effect on cross-border e-commerce trade. Furthermore, the results suggest that deviation from a cointegration system of cross-border e-commerce trade and international logistics will lead to the cross-border e-commerce trade and international logistics changing within the range of approximately 2.2% to 47.2% in the next period. Therefore, referring to these findings, each OECD country's government should take up corresponding policies to ensure the sustainable development of both international logistics and cross-border e-commerce trade.

**Keywords:** international logistics; cross-border e-commerce trade; panel vector error correction model; sustainable development

## 1. Introduction

Cross-border e-commerce trade brings a market for the development of international logistics. Traditional business models are increasingly unable to meet the needs of people. In the new era, consumers will pay more attention to both the quality and variety of goods. In addition, consumers will also focus on the shopping experience. Fortunately, the emergence of cross-border e-commerce trade will greatly enhance the shopping convenience, meet consumer demand, and optimize the consumer shopping experience. Meanwhile, the emergence of cross-border e-commerce trade also plays a role in improving the quality of service, improving the effectiveness of the supply chain, enhancing the efficiency of business operations and increasing the volume of international trade. Therefore, many traditional enterprises have introduced the model of cross-border e-commerce trade. The huge cross-border e-commerce trade market provides market opportunities for the development of international logistics and a necessary link for cross-border e-commerce trade. On the contrary, international logistics is an essential part of building a cross-border e-commerce trade supply chain. The process of cross-border e-commerce trade includes negotiation, contracting, payment, logistics and other aspects. The development of cross-border e-commerce trade also provides market opportunities for the development of enterprises related to these links. In cross-border e-commerce trade, the basis of corporate and consumer contract practice is non-virtual "international logistics". The factors affecting consumer consumption experience also include the efficiency and cost

of logistics. Therefore, cross-border e-commerce trade not only provides market opportunities for the development of international logistics, but also challenges its development. As such, international logistics is an indispensable link in the development of cross-border e-commerce trade. Its development level has also become a key factor in the integration of the cross-border e-commerce trade supply chain.

In reality, the sustainable development of both international logistics and cross-border e-commerce trade also faces many challenges. From the theoretical level, the rapid development of cross-border e-commerce trade can increase transaction orders, which can promote the sustainable development of international logistics. However, in the actual operation, the coordination and sustainability of both international logistics and cross-border e-commerce trade are fragile. As we know, the e-commerce platform uses the internet to establish communication links with consumers, which provides consumers with more convenient product purchase channels. Transactions can be achieved only with the help of the network, which is not limited by both space and time. As a platform transportation partner, international logistics needs to do a good job in customs, such as in quality inspection and other work, after receiving the international logistics transportation requirements sent by the platform. As a matter of fact, each link is very complex and takes a lot of time. To solve the time-consuming problem, some cross-border e-commerce trade service providers choose to establish an overseas storage model to offer customers return and exchange services. Due to the large cost of overseas warehousing construction, the demand for product sales data information is high. At present, the international logistics enterprises cannot obtain the product warehouse management data information, which leads to the low accuracy of warehouse design. The layout of storage space cannot meet the actual storage needs, which is not conducive to the improvement of warehouse management level, and international logistics enterprises have always struggled to maintain sustainable development. However, the cross-border e-commerce platform does not know enough about overseas warehousing, and does not understand the importance of international logistics enterprises to their own sustainable development. As a result, the sustainability of bilateral cooperation is weak as a whole.

Furthermore, to improve the service, the cross-border e-commerce enterprises need to cooperate with international logistics enterprises to provide high-quality services for consumers. This operation goal needs to be completed by both parties' synchronous operation. However, in the actual operation process, the cross-border e-commerce enterprises focus on their own work. To let consumers experience high-quality shopping services, they do not consider whether the work of international logistics enterprises is difficult or not. In the international logistics transportation services, the cross-border e-commerce enterprises will set the cooperation requirements of the nearest warehouse delivery, which limits the sustainable operation of international logistics enterprises. Considering the feasibility of international logistics delivery, human resources, transportation costs and other issues, international logistics enterprises are unable to meet the requirements of this international distribution, which results in contradictions in the operation and cooperation between both the parties.

In addition, there are still many problems worth discussing between cross-border e-commerce trade and international logistics. Therefore, a larger number of scholars in this field have made attempts to study this proposition. Ying [1] performed research into logistics mode selection in B2C cross-border e-commerce in China. His findings show that the lagged construction of international logistics systems and the high cost of international logistics have seriously hindered the further development of China's B2C cross-border e-commerce enterprises. With different samples such as India, Alyoubi [2] found that the international logistics problems pose a significant barrier for the sustainable development of e-commerce trade. However, Wang and Lee [3] set China as an example to discuss the effect of international logistics on international trade. They found that this kind of effect is positive. Meanwhile, Cho et al. [4] also found that the international logistics capability is positively correlated with the enterprises' performance in terms of cross-border

e-commerce market. To this day, although a great many experts have deeply explored the dynamic relationship between cross-border e-commerce trade and international logistics, they have not reached a consensus. As a matter of fact, due to different perspectives, different samples, different times spans and different research approaches, drawing different conclusions about this proposition can be understood. Said differently, it is highly valuable to exploit the relationship between cross-border e-commerce trade and international logistics. Because of this background, this paper sets Organization for Economic Co-operation and Development (OECD) countries as an example over the period 2000–2018 to explore the dynamic relationship between international logistics and cross-border e-commerce trade. Then, the panel data will be used to carry out an empirical analysis under some econometric approaches, such as the panel unit root test, the panel cointegration test, the panel causality test and the panel vector error correction model. Moreover, there are seven variables used in this paper. They are the cross-border e-commerce export trade, the cross-border e-commerce import trade, the GDP per capita, the population, the relative price, the international logistics, and the one-period lagged outward foreign direct investment. Finally, all these variables and econometric methods will be employed to discuss the dynamic relationship between cross-border e-commerce trade and international logistics.

The contribution of this current paper to the previous empirical literature focusing on the relationship between cross-border e-commerce trade and international logistics is fourfold. First, this paper attempts to fill the gap in the empirical literature with regard to the dynamic relationship between cross-border e-commerce trade and international logistics in the Organization for Economic Co-operation and Development (OECD) countries. The examination with the OECD countries regarded as a case study is particularly interesting, since most of these countries are sufficiently open to each other and fully integrated into one economic market to have a high potential for performing economic activities in both new trade modes and new logistics modes. Meanwhile, most of them have a huge capacity for producing a great quantity of goods and platforms to conduct cross-border e-commerce trade. Moreover, most of them also have extremely advanced international logistics systems to satisfy the needs of cross-border e-commerce trade activities. These fundamentals make our results more reliable and robust. Second, unlike previous research that has used traditional econometric models, such as the vector auto-regression model and the vector error correction model, this paper employs a recent multivariate economic technique which is famous as the panel data vector error correction model. Because the properties of both the time series and the cross-section of the data are taken into consideration, this can help us to more accurately estimate the long-run and short-run relationship between cross-border e-commerce trade and international logistics. Thirdly, a large number of panel unit root tests are used to test the stationarity of variables used in this paper. This can also ensure the accuracy of the empirical results. Finally, the evidence this paper provides can form a foundation for future scholars who are interested in exploring the dynamic relationship between cross-border e-commerce trade and international logistics.

To this end, the rest of this paper will proceed as follows. Chapter two discusses the previous research and develops hypotheses. Chapter three describes the variables and analyzes model specifications. Chapter four presents the results and discussions. Chapter five draws the conclusions, puts forward some corresponding suggestions and presents limitations.

## 2. Literature Review and Hypothesis Development

Since the data acquisition of cross-border e-commerce trade is limited, the empirical research on this issue is still at an initial stage. Different studies draw different conclusions. In other words, the relationship between international logistics and cross-border e-commerce trade is not fully confirmed.

### 2.1. Sustainable Development of Cross-Border E-Commerce Trade and Hypothesis

With respect to the factors that can influence the sustainable development of cross-border e-commerce trade, some previous research should be mentioned. Gong et al. [5] found that both sustainable supply chain performance and management can expand the volume of cross-border e-commerce trade. Meanwhile, Zimon et al. [6] also agreed with this results. Moreover, with different samples and approaches, Zimon et al. [7] agreed with this view. However, in this area, more emphasis will be laid on the effect of international logistics on the sustainable development of cross-border e-commerce trade. From the East Asian perspective, Carruthers et al. [8] studied the relationship between trade and logistics. They found that reducing the cost and improving the quality of logistics and transport systems can improve international market access and can directly lead to an increase in the trade. Meanwhile, Hausman et al. [9] performed research on the impact of logistics performance on global bilateral trade. Their findings indicate that the improvement of logistics performance can increase bilateral trade, with a sample of 80 countries from the World Bank Group. Moreover, Anderson and Villa [10] set the North American Free Trade Agreement (NAFTA) region as a sample to discuss the relationship between cross-border transportation and trade across international borders. Using a fixed effect model, they found that cross-border transportation is an important factor that affects the trade across international borders. Furthermore, it is apparent that geographical factors and international logistics are two important determinants of cross-border e-commerce trade competitiveness. Stated differently, the distribution and capability of a country's international logistics facilities play an increasingly important role in the design of a business strategy that is aimed at improving a country's market share in cross-border e-commerce trade. Due to this background, Bensassi et al. [11] tried to use the augmented gravity model of trade, including international logistics as the explanatory variable, with a sample of 19 Spanish regions associated with 64 destinations from 2003 to 2007. They found that international logistics is very important for the analysis of cross-border e-commerce trade flows. They emphasized the importance of logistics measures at the regional level. They found that the quantity, scale and quality of international logistics facilities has a positive impact on cross-border e-commerce export flows. However, Hesse [12] found that there is a negative effect of both logistics and freight transport on e-commerce trade due to inefficient physical distribution in the short-run.

In addition, Hsiao et al. [13] found that good cross-border logistics is a driving factor in promoting cross-border e-commerce trade. However, the weakness and shortage of the international logistics function in inland areas have shaped a bottleneck, which restricts the development of cross-border e-commerce trade. Based on the One Belt and One Road Initiative, the development of cross-border e-commerce demand in inland regions is provoking the demand for inland international logistics network functions. The dry ports have played a vital role in constructing international logistics networks, which has obtained more attention from inland governments. Gani [14] attempted to exploit the function of international logistics performance in cross-border e-commerce trade. He found that the overall international logistics performance has a significantly positive effect on cross-border e-commerce trade statistics in the long run. He also expanded his analysis by investigating whether the particularity of logistics is important to cross-border e-commerce trade. From this expansion analysis, he found that obtaining good international logistics performance in many dimensions has statistical significance and a positive effect on cross-border e-commerce export trade. Said differently, sustainable investment in international logistics infrastructure and services can have a positive effect on cross-border e-commerce trade.

Why does cross-border e-commerce trade flourish in some countries while other countries fail to develop it? This common problem is often mentioned in the literature. Halaszovich and Kinra [15] treated Asian regions as an example to explore the deep mechanism of this issue. Their study provided some theoretical evidence that good international logistics can offset some of the negative effects of distance on cross-border e-commerce trade in the long-run. Moreover, they also found that international logistics in more devel-

oped countries can more easily overcome the distance costs to some extent in the short-run. In fact, this conclusion was also found by Refs. [16,17]. Wei and Dong [18] set China as an example to study the same proposition. They found that a satisfactory port international logistics foundation can provide a good condition for cross-border e-commerce trade expansion. The sustainable growth of cross-border e-commerce trade depends on the efficiency of a trade support structure, such as the international logistics service. Although logistics plays an indispensable role in supporting business activities, both the number of practitioners of trade analysis and the focus on trade policy research are generally low. Along with the economic globalization and network generalization, this provides a good opportunity for the development of cross-border e-commerce trade.

To this end, based on the empirical analyses above, the hypothesis can be derived as follows:

**Hypothesis 1 (H1):** *International logistics is positively related to the sustainable development of cross-border e-commerce trade in the long-run.*

**Hypothesis 2 (H2):** *International logistics is negatively related to the sustainable development of cross-border e-commerce trade in the short-run.*

*2.2. Sustainable Development of International Logistics and Hypothesis*

In this section, more attention will be paid to the effect of cross-border e-commerce trade on the sustainable development of international logistics. Delfmann et al. [19] found that e-commerce trade is favored by logistics service providers, which is beneficial for the sustainable development of logistics. This finding is also verified by Ref. [20]. Leinbach [21] set North America as a case to discuss this issue. He found that globalized e-commerce really leads to a sustainable growth of international logistics. Meanwhile, Nguyen and Tongzon [22] regarded Australia and China as a study subject to explore the causal relationship between cross-border e-commerce trade and the development of Australian logistics based on the vector auto-correlation framework. They found that the growth of cross-border e-commerce trade between Australia and China has led to the development of logistics in Australia. Żurek [23] found that due to the sustainable development of e-commerce trade, a new logistics chain management strategy has started to show up, which contains processes of handling both offline and online sales channels. Therefore, the sustainable development of logistics will be changed. Based on this research, Qin et al. [23] also support this idea.

Kadłubek [24] has attempted to study the relationship between e-commerce trade and e-logistics with the case of Poland. In the long-run, e-commerce trade can positively expand the scale of e-logistics. Schöder [25] attempted to find some sustainable development solutions for urban logistics. In the short-run, the increase in freight volume caused by the development of e-commerce brings challenges to urban logistics. However, in the long-run, the improvement and rapid development of e-commerce trade platforms also provides a guarantee for the sustainability of urban logistics. Hong et al. [26] set Korea as an example to study logistics with the Association of Southeast Asian Nations. They found that both the increase in cross-border e-commerce platform and the expansion of e-commerce trade volume bring positive effects for the sustainable development of logistics between both two parties. In countries such as Mongolia and Georgia, Wang et al. [27] found that without long-term economic and social development, national e-commerce trade cannot fully support the sustainable development of logistics performance.

To sum up, based on the empirical analyses above, hypotheses can be derived, as follows:

**Hypothesis 3 (H3):** *Cross-border e-commerce trade is positively related to the sustainable development of international logistics in the long-run;*

**Hypothesis 4 (H4):** *Cross-border e-commerce trade is negatively related to the sustainable development of international logistics in the short-run.*

To summarize, these papers analyzed above have debated the relationship between international logistics and cross-border e-commerce trade with different frameworks and different perspectives. In order to make a different contribution from others, this paper sets OECD countries as an example to explore the dynamic relationship between international logistics and cross-border e-commerce trade under a series of econometric approaches, such as the panel unit root test, the panel vector error correction model, and so on. In fact, this paper enriches the existing literature in general, in terms of the findings and research methodology.

## 3. Theoretical Framework

### 3.1. Model Specification

To estimate the dynamic relationship between international logistics and cross-border e-commerce trade, we set up four basic models, as the following indicates.

The long-run effect of international logistics on cross-border e-commerce export trade is modeled in Equation (1):

$$\log ecex_{i,t} = c_{1i} + \alpha_1 \log il_{i,t} + \alpha_2 \log dgdp_{i,t} \\ + \alpha_3 \log pop_{i,t} + \alpha_4 \log rp_{i,t} + \alpha_5 \log ofdi_{i,t-1} + \mu_{1i,t} \tag{1}$$

The long-run effect of cross-border e-commerce export trade on international logistics is modeled in Equation (2):

$$\log il_{i,t} = c_{2i} + \beta_1 ecex_{i,t} + \beta_2 \log dgdp_{i,t} \\ + \beta_3 \log pop_{i,t} + \beta_4 \log rp_{i,t} + \beta_5 \log ofdi_{i,t-1} + \mu_{2i,t} \tag{2}$$

The long-run effect of international logistics on cross-border e-commerce import trade is modeled in Equation (3):

$$\log ecim_{i,t} = c_{3i} + \gamma_1 \log il_{i,t} + \gamma_2 \log dgdp_{i,t} \\ + \gamma_3 \log pop_{i,t} + \gamma_4 \log rp_{i,t} + \gamma_5 \log ofdi_{i,t-1} + \mu_{3i,t} \tag{3}$$

The long-run effect of cross-border e-commerce import trade on international logistics is modeled in Equation (4):

$$\log il_{i,t} = c_{4i} + \delta_1 ecim_{i,t} + \delta_2 \log dgdp_{i,t} \\ + \delta_3 \log pop_{i,t} + \delta_4 \log rp_{i,t} + \delta_5 \log ofdi_{i,t-1} + \mu_{4i,t} \tag{4}$$

where $\log ecex_{i,t}$ represents the cross-border e-commerce export trade (it is defined as the total volume of cross-border e-commerce export trade); $\log ecim_{i,t}$ represents the cross-border e-commerce import trade (it is defined as the total volume of cross-border e-commerce import trade); $\log il_{i,t}$ represents the international logistics (it is defined as the international freight transport in million tons per kilometer); $\log dgdp_{i,t}$ represents the GDP per capita (at the constant 2000 price); $\log pop_{i,t}$ represents the population; $\log rp_{i,t}$ represents the relative price level; $\log ofdi_{i,t-1}$ represents the one-period lagged outward foreign direct investment. $c_{1i}, c_{2i}, c_{3i}, c_{4i}$ and $c_{5i}$ represent the fixed effects and the heterogeneity among cross-sections; $i$ and $t$ represent the country and year, respectively. Moreover, log represents the logarithmic equation. $\alpha_i, \beta_i, \gamma_i$ and $\delta_i$ ($i = 1, 2, 3, 4$) represent those coefficients of each variable. $u_{1i,t}, u_{2i,t}, u_{3i,t}$ and $u_{4i,t}$ represent independent and identical distributions among countries and years.

The GDP per capita, the population, the relative price, and the one-period lagged outward foreign direct investment are always treated as important factors that affect the cross-border e-commerce trade in previous research. Likewise, these variables are also introduced in this paper. In detail, an increase in the GDP per capita indicates a high dis-

posable income. However, a high disposable income will spur the consumer's purchasing desire, which will affect the cross-border e-commerce trade. The large population size, according to the production function, means more output. However, a large amount of output will stimulate cross-border e-commerce trade. The relative price level is introduced into this paper as a positive difference between two countries' price levels will yield additional benefits in terms of the home country. Of course, because of this advantage, the home country is willing to fulfil the cross-border e-commerce trade. The outward foreign direct investment has a certain impact on the productivity of the host country's domestic enterprises. The competitive effect of foreign capital injection will occupy the market of domestic enterprises. It will also help the host country's enterprises to improve their productivity, through the technology spillover effect of foreign capital enterprises on a host country's enterprises. This comprehensive effect will affect the productivity of enterprises, and thus affect the export trade of the whole country, and its influence depends on the magnitude of the role is played. In addition, in the process of globalization, the growth of foreign trade also affects the investment decisions of foreign investors as regards the host country. As such, in order to avoid the endogenous problems caused by the two-way causality between outward foreign direct investment and export trade, one-period lagged outward foreign direct investment is introduced into this paper. This paper selects the OECD countries as an example to explore the dynamic relationship between international logistics and cross-border e-commerce trade. All these panel data employed in this paper are sourced from the World Bank, the Data Center of Organization for Economic Cooperation and Development, the United Nations Conference on Trade and Development, Statista and iimedia.

As for model (1), model (2), model (3), and model (4), we assume that all variables are generated by the panel unit root process, and the residuals of model (1), model (2), model (3) and model (4) are stationary processes. Stated differently, $u_{1i,t}$, $u_{2i,t}$, $u_{3i,t}$ and $u_{4i,t}$ are independently and identically distributed among countries and years. Therefore, it can be concluded that model (1), model (2), model (3), and model (4) are a set of panel cointegration models. As such, the panel vector error correction models (short-run effect) are presented as shown below.

The short-run effect of international logistics on cross-border e-commerce export trade is modeled in Equation (5):

$$\Delta \log ecex_{i,t} = \lambda_1 ecm_{i,t-k-1} + \phi_1 \Delta \log il_{i,t-k} + \phi_2 \Delta \log dgdp_{i,t-k} \\ + \phi_3 \Delta \log pop_{i,t-k} + \phi_4 \Delta \log rp_{i,t-k} + \phi_5 \Delta \log ofdi_{i,t-k-1} + \varepsilon_{1i,t} \tag{5}$$

The short-run effect of cross-border e-commerce export trade on international logistics is modeled in Equation (6):

$$\Delta \log il_{i,t} = \lambda_2 ecm_{i,t-k-1} + \varphi_1 \Delta \log ecex_{i,t-k} + \varphi_2 \Delta \log dgdp_{i,t-k} \\ + \varphi_3 \Delta \log pop_{i,t-k} + \varphi_4 \Delta \log rp_{i,t-k} + \varphi_5 \Delta \log ofdi_{i,t-k-1} + \varepsilon_{2i,t} \tag{6}$$

The short-run effect of international logistics on cross-border e-commerce import trade is modeled in Equation (7):

$$\Delta \log ecim_{i,t} = \lambda_3 ecm_{i,t-k-1} + \eta_1 \Delta \log il_{i,t-k} + \eta_2 \Delta \log dgdp_{i,t-k} \\ + \eta_3 \Delta \log pop_{i,t-k} + \eta_4 \Delta \log rp_{i,t-k} + \eta_5 \Delta \log ofdi_{i,t-k-1} + \varepsilon_{3i,t} \tag{7}$$

The short-run effect of cross-border e-commerce import trade on international logistics is modeled in Equation (8):

$$\Delta \log il_{i,t} = \lambda_4 ecm_{i,t-k-1} + \kappa_1 \Delta \log ecim_{i,t-k} + \kappa_2 \Delta \log dgdp_{i,t-k} \\ + \kappa_3 \Delta \log pop_{i,t-k} + \kappa_4 \Delta \log rp_{i,t-k} + \kappa_5 \Delta \log ofdi_{i,t-k-1} + \varepsilon_{4i,t} \tag{8}$$

where $ecm_{i,t-k-1}$ represents the residual of panel cointegration models; $\lambda_1, \lambda_2, \lambda_3$ and $\lambda_4$ represent the short-term adjustment effects. In other words, these coefficients represent the

effects of a long-term relationship among international logistics, cross-border e-commerce export trade, cross-border e-commerce import trade, GDP per capita, population, relative price level, and one-period lagged outward foreign direct investment, on the short-term changes in every variable. $\Delta$ represents the first difference operator. $\phi_i, \varphi_i, \eta_i$ and $\kappa_i$ ($i = 1, 2, 3, 4$) represent the coefficients of each variable. $\varepsilon_{1i,t}, \varepsilon_{2i,t}, \varepsilon_{3i,t}$ and $\varepsilon_{4i,t}$ represent independent and identical distribution among countries and years. Additionally, when $\lambda_1$ is less than zero, it means that the long-term relationship suppresses changes in cross-border e-commerce export trade in the short term. From the perspective of the econometric approach, this verifies that model (1) is a panel cointegration model. When $\lambda_2$ is less than zero, it means that the long-term relationship suppresses changes in international logistics in the short term. From the perspective of the econometric approach, this proves that model (2) is a panel cointegration model. When $\lambda_3$ is less than zero, it means that the long-term relationship suppresses changes in cross-border e-commerce import trade in the short term. From the perspective of the econometric approach, this testifies that model (3) is a panel cointegration model. When $\lambda_4$ is less than zero, it means that the long-term relationship suppresses changes in international logistics in the short term. From the perspective of the econometric approach, this indicates that model (4) is a panel cointegration model.

### 3.2. Panel Unit Root Test

To avoid the occurrence of pseudo-regression and other problems, according to the structural characteristics of the data, the panel unit root test will be used to examine the stationarity of variables before conducting further estimations. Clearly, common unit root test methods have often failed in unit root tests of panel data. The robustness of the results must be guaranteed when the panel vector error correction analyses are carried out. In this paper, the IPS test, Fisher test, PP test, LLC test and other methods are used to test the stationarity of each variable, respectively. The test process will be kept in line with the models below.

Based on the research of Ref. [28], the LLC test model gives

$$\Delta y_{i,t} = \rho_i y_{i,t-1} + \sum_{j=1}^{k} \gamma_{i,j}\Delta y_{i,t-j} + X'_{i,t}\beta_{i,t} + \mu_{i,t} \tag{9}$$

where $X'_{i,t}$ indicates the exogenous variables.

Based on the research of Ref. [29], the IPS test model gives

$$\Gamma_t = \left( \frac{\sqrt{N}[t_{NT}(p) - a_{NT}]}{\sqrt{b_{NT}}} \right) \to N(0,1) \tag{10}$$

where $a_{NT} = \left(\frac{1}{N}\right)\sum_{i=1}^{N} E[t_{NT}(p,0)]$ and $b_{NT} = \left(\frac{1}{N}\right)\sum_{i=1}^{N} \text{var}[t_{NT}(p,0)] \cdot t_{NT}$ represents the ADF statistics with the lag $p$ of the $N$ cross-sectional individuals. $E[t_{NT}(p,0)]$ represents the ADF statistic mean with the lag $p$ of $N$ cross-sectional individuals. $\text{var}[t_{NT}(p,0)]$ indicates the ADF statistic variance with the lag $p$ of $N$ cross-sectional individuals.

Based on the research of Ref. [30], the Hadri test model gives

$$LM_1 = \frac{1}{N}\left( \sum_{I=1}^{N} (\sum_{T} S_i(t)^2 / T_2) / \overline{f}_0 \right) \tag{11}$$

$$LM_2 = \frac{1}{N}\left( \sum_{I=1}^{N} (\sum_{T} S_i(t)^2 / T^2) / f_{i0} \right) \tag{12}$$

where $S_i(t)^2 = \sum_{s=1}^{t} \hat{\varepsilon}_{i,t}$ represents the sum of residuals, and $\overline{f}_0 = \sum_{t=1}^{n} \frac{f_{i0}}{n}$ represents the individual mean.

Based on the research of Refs. [31,32], the Fisher-ADF and Fisher-PP test model gives:

$$\chi^2 = -2\sum_{i=1}^{N} \log(p_i) \tag{13}$$

where $p_i$ indicates the corresponding $p$-value of the ADF test and PP test, respectively.

### 3.3. Panel Cointegration Test

Engle and Granger [33] found that the determination of the long-term relationship between variables indicates the adjustment period of the causal relationship between related variables. If the variables have unit roots at levels (namely, they are non-stationary), they need to be taken as the first difference. After the first difference, these variables do not have a unit root (namely, they are stationary). In other words, these variables are cointegrated at $I(1)$. We assume that there is a unidirectional relationship between cross-border e-commerce export trade, cross-border e-commerce, GDP per capita, population, relative price level and one-period lagged outward foreign direct investment. For example, for model (1), international logistics granger causes cross-border e-commerce export trade. For model (2), cross-border e-commerce export trade granger causes international logistics. For model (3), international logistics granger causes cross-border e-commerce import trade. For model (4), cross-border e-commerce import trade granger causes international logistics. In this paper, the combined Johansen–Fisher panel cointegration approach will be employed to determine the long-run equilibrium in international logistics (model (1)) in relation to cross-border e-commerce export trade; cross-border e-commerce export trade (model (2)) in relation to international logistics; international logistics (model (3)) in relation to cross-border e-commerce import trade, and cross-border e-commerce import trade (model (4)) in relation to international logistics. The combined Johansen–Fisher system procedure for the long-term equilibrium of these variables obeys the panel vector auto-regressive system equations. These system equations give

$$\Delta Y_{i,t} = \sum_{i=1}^{k} \Gamma_{i,t} \Delta Y_{i,t-j} + v_{i,t} \tag{14}$$

For model (1), $Y_{\alpha i,t} = \begin{bmatrix} \log ecex_{i,t} \\ \log il_{i,t} \\ \log dgdp_{i,t} \\ \log pop_{i,t} \\ \log rp_{i,t} \\ \log ofdi_{i,t} \end{bmatrix}$, $\Gamma_{\alpha i,t} = \begin{bmatrix} \Gamma_{11i,t}, \Gamma_{12i,t}, \Gamma_{13i,t}, \Gamma_{14i,t}, \Gamma_{15i,t}, \Gamma_{16i,t} \\ \Gamma_{21i,t}, \Gamma_{22i,t}, \Gamma_{23i,t}, \Gamma_{24i,t}, \Gamma_{25i,t}, \Gamma_{26i,t} \\ \Gamma_{31i,t}, \Gamma_{32i,t}, \Gamma_{33i,t}, \Gamma_{34i,t}, \Gamma_{35i,t}, \Gamma_{36i,t} \\ \Gamma_{41i,t}, \Gamma_{42i,t}, \Gamma_{43i,t}, \Gamma_{44i,t}, \Gamma_{45i,t}, \Gamma_{46i,t} \\ \Gamma_{51i,t}, \Gamma_{52i,t}, \Gamma_{53i,t}, \Gamma_{54i,t}, \Gamma_{55i,t}, \Gamma_{56i,t} \\ \Gamma_{61i,t}, \Gamma_{62i,t}, \Gamma_{63i,t}, \Gamma_{64i,t}, \Gamma_{65i,t}, \Gamma_{66i,t} \end{bmatrix}$

and $v_{\alpha i,t} = \begin{bmatrix} v_{1i,t} \\ v_{2i,t} \\ v_{3i,t} \\ v_{4i,t} \\ v_{5i,t} \\ v_{6i,t} \end{bmatrix}$. For model (2), $Y_{\beta i,t} = \begin{bmatrix} \log il_{i,t} \\ \log ecex_{i,t} \\ \log dgdp_{i,t} \\ \log pop_{i,t} \\ \log rp_{i,t} \\ \log ofdi_{i,t} \end{bmatrix}$,

$\Gamma_{\beta i,t} = \begin{bmatrix} \Gamma_{11i,t}, \Gamma_{12i,t}, \Gamma_{13i,t}, \Gamma_{14i,t}, \Gamma_{15i,t}, \Gamma_{16i,t} \\ \Gamma_{21i,t}, \Gamma_{22i,t}, \Gamma_{23i,t}, \Gamma_{24i,t}, \Gamma_{25i,t}, \Gamma_{26i,t} \\ \Gamma_{31i,t}, \Gamma_{32i,t}, \Gamma_{33i,t}, \Gamma_{34i,t}, \Gamma_{35i,t}, \Gamma_{36i,t} \\ \Gamma_{41i,t}, \Gamma_{42i,t}, \Gamma_{43i,t}, \Gamma_{44i,t}, \Gamma_{45i,t}, \Gamma_{46i,t} \\ \Gamma_{51i,t}, \Gamma_{52i,t}, \Gamma_{53i,t}, \Gamma_{54i,t}, \Gamma_{55i,t}, \Gamma_{56i,t} \\ \Gamma_{61i,t}, \Gamma_{62i,t}, \Gamma_{63i,t}, \Gamma_{64i,t}, \Gamma_{65i,t}, \Gamma_{66i,t} \end{bmatrix}$ and $v_{\beta i,t} = \begin{bmatrix} v_{1i,t} \\ v_{2i,t} \\ v_{3i,t} \\ v_{4i,t} \\ v_{5i,t} \\ v_{6i,t} \end{bmatrix}$. For model (3),

$$Y_{\gamma i,t} = \begin{bmatrix} \log ecim_{i,t} \\ \log il_{i,t} \\ \log dgdp_{i,t} \\ \log pop_{i,t} \\ \log rp_{i,t} \\ \log ofdi_{i,t} \end{bmatrix}, \quad \Gamma_{\gamma i,t} = \begin{bmatrix} \Gamma_{11i,t}, \Gamma_{12i,t}, \Gamma_{13i,t}, \Gamma_{14i,t}, \Gamma_{15i,t}, \Gamma_{16i,t} \\ \Gamma_{21i,t}, \Gamma_{22i,t}, \Gamma_{23i,t}, \Gamma_{24i,t}, \Gamma_{25i,t}, \Gamma_{26i,t} \\ \Gamma_{31i,t}, \Gamma_{32i,t}, \Gamma_{33i,t}, \Gamma_{34i,t}, \Gamma_{35i,t}, \Gamma_{36i,t} \\ \Gamma_{41i,t}, \Gamma_{42i,t}, \Gamma_{43i,t}, \Gamma_{44i,t}, \Gamma_{45i,t}, \Gamma_{46i,t} \\ \Gamma_{51i,t}, \Gamma_{52i,t}, \Gamma_{53i,t}, \Gamma_{54i,t}, \Gamma_{55i,t}, \Gamma_{56i,t} \\ \Gamma_{61i,t}, \Gamma_{62i,t}, \Gamma_{63i,t}, \Gamma_{64i,t}, \Gamma_{65i,t}, \Gamma_{66i,t} \end{bmatrix} \quad \text{and}$$

$$v_{\gamma i,t} = \begin{bmatrix} v_{1i,t} \\ v_{2i,t} \\ v_{3i,t} \\ v_{4i,t} \\ v_{5i,t} \\ v_{6i,t} \end{bmatrix}. \quad \text{For model (4),} \quad Y_{\delta i,t} = \begin{bmatrix} \log il_{i,t} \\ \log ecim_{i,t} \\ \log dgdp_{i,t} \\ \log pop_{i,t} \\ \log rp_{i,t} \\ \log ofdi_{i,t} \end{bmatrix},$$

$$\Gamma_{\delta i,t} = \begin{bmatrix} \Gamma_{11i,t}, \Gamma_{12i,t}, \Gamma_{13i,t}, \Gamma_{14i,t}, \Gamma_{15i,t}, \Gamma_{16i,t} \\ \Gamma_{21i,t}, \Gamma_{22i,t}, \Gamma_{23i,t}, \Gamma_{24i,t}, \Gamma_{25i,t}, \Gamma_{26i,t} \\ \Gamma_{31i,t}, \Gamma_{32i,t}, \Gamma_{33i,t}, \Gamma_{34i,t}, \Gamma_{35i,t}, \Gamma_{36i,t} \\ \Gamma_{41i,t}, \Gamma_{42i,t}, \Gamma_{43i,t}, \Gamma_{44i,t}, \Gamma_{45i,t}, \Gamma_{46i,t} \\ \Gamma_{51i,t}, \Gamma_{52i,t}, \Gamma_{53i,t}, \Gamma_{54i,t}, \Gamma_{55i,t}, \Gamma_{56i,t} \\ \Gamma_{61i,t}, \Gamma_{62i,t}, \Gamma_{63i,t}, \Gamma_{64i,t}, \Gamma_{65i,t}, \Gamma_{66i,t} \end{bmatrix} \quad \text{and} \quad v_{\delta i,t} = \begin{bmatrix} v_{1i,t} \\ v_{2i,t} \\ v_{3i,t} \\ v_{4i,t} \\ v_{5i,t} \\ v_{6i,t} \end{bmatrix}. \quad \text{For the convenience}$$

of writing, we assume that $ofdi_{t-1}$ is equal to $ofdi_t$. The reduced form of these matrices as set out by Enders (2008) gives

$$\log ecex_{i,t} = [\log il_{i,t}, \log dgdp_{i,t}, \log pop_{i,t}, \log rp_{i,t}, \log ofdi_{i,t-1}] \tag{15}$$

$$\log il_{i,t} = [\log ecex_{i,t}, \log dgdp_{i,t}, \log pop_{i,t}, \log rp_{i,t}, \log ofdi_{i,t-1}] \tag{16}$$

$$\log ecim_{i,t} = [\log il_{i,t}, \log dgdp_{i,t}, \log pop_{i,t}, \log rp_{i,t}, \log ofdi_{i,t-1}] \tag{17}$$

$$\log il_{i,t} = [\log ecim_{i,t}, \log dgdp_{i,t}, \log pop_{i,t}, \log rp_{i,t}, \log ofdi_{i,t-1}] \tag{18}$$

*3.4. Panel Causality Test*

The panel cointegration test only exhibits the long-term relationship among cross-border e-commerce export trade, cross-border e-commerce import trade, GDP per capita, population, relative price level, international logistics, and one-period lagged outward foreign direct investment. However, it has not explained how these variables affect each other. Namely, the direction of causality needs to be identified. For this problem, Dumitrescu and Hurlin [34] developed a panel causality test approach to reveal the direction of causality among these variables used in this paper. It is a dynamic panel test, which is called the pairwise Dumitrescu–Hurlin test. Compared with other approaches, it is more efficient and robust on the basis of estimation. The models of the pairwise Dumitrescu–Hurlin panel causality tests give

$$\Delta \log ecex_{i,t} = \theta_{1j} + \varsigma_{11ik} \sum_{k=1}^{p1} \Delta \log il_{i,t-k} + \varsigma_{12ik} \sum_{k=1}^{p2} \Delta \log dgdp_{i,t-k}$$
$$+ \varsigma_{13ik} \sum_{k=1}^{p3} \Delta \log pop_{i,t-k} + \varsigma_{14ik} \sum_{k=1}^{p4} \Delta \log rp_{i,t-k} \tag{19}$$
$$+ \varsigma_{15ik} \sum_{k=1}^{p5} \Delta \log ofdi_{i,t-k-1} + v_{1i,t}$$

$$\Delta \log il_{i,t} = \theta_{2j} + \tau_{11ik} \sum_{k=1}^{p1} \Delta \log ecex_{i,t-k} + \tau_{12ik} \sum_{k=1}^{p2} \Delta \log dgdp_{i,t-k}$$
$$+ \tau_{13ik} \sum_{k=1}^{p3} \Delta \log pop_{i,t-k} + \tau_{14ik} \sum_{k=1}^{p4} \Delta \log rp_{i,t-k} \tag{20}$$
$$+ \tau_{15ik} \sum_{k=1}^{p5} \Delta \log ofdi_{i,t-k-1} + v_{2i,t}$$

$$\Delta \log ecim_{i,t} = \theta_{3j} + \xi_{11ik} \sum_{k=1}^{p1} \Delta \log il_{i,t-k} + \xi_{12ik} \sum_{k=1}^{p2} \Delta \log dgdp_{i,t-k}$$
$$+\xi_{13ik} \sum_{k=1}^{p3} \Delta \log pop_{i,t-k} + \xi_{14ik} \sum_{k=1}^{p4} \Delta \log rp_{i,t-k} \qquad (21)$$
$$+\xi_{15ik} \sum_{k=1}^{p5} \Delta \log ofdi_{i,t-k-1} + v_{3i,t}$$

$$\Delta \log il_{i,t} = \theta_{4j} + \psi_{11ik} \sum_{k=1}^{p1} \Delta \log ecim_{i,t-k} + \psi_{12ik} \sum_{k=1}^{p2} \Delta \log dgdp_{i,t-k}$$
$$+\psi_{13ik} \sum_{k=1}^{p3} \Delta \log pop_{i,t-k} + \psi_{14ik} \sum_{k=1}^{p4} \Delta \log rp_{i,t-k} \qquad (22)$$
$$+\psi_{15ik} \sum_{k=1}^{p5} \Delta \log ofdi_{i,t-k-1} + v_{4i,t}$$

where $\Delta$ represents the first difference operator. $p_1$, $p_2$, $p_3$, $p_4$ and $p_5$ represent the lag length. $v_{1i,t}$, $v_{2i,t}$, $v_{3i,t}$ and $v_{4i,t}$ represent independent and identical distributions. $i$ represents the country. $t$ represents the year.

## 4. Empirical Analyses

### 4.1. Summary Statistics

In this paper, seven variables will be used. They include the cross-border e-commerce export trade ($\log ecex_{i,t}$), cross-border e-commerce import trade ($\log ecim_{i,t}$), GDP per capita ($\log pgdp_{i,t}$), population ($\log pop_{i,t}$), relative price level ($\log rp_{i,t}$), international logistics ($\log il_{i,t}$) and one-period lagged outward foreign direct investment ($\log ofdi_{i,t-1}$). Both the summary statistics and the OECD countries list will be shown in Tables 1 and 2.

**Table 1.** Summary Statistic.

| Variable | Observation | Mean | Standard Deviation | Min | Max |
|---|---|---|---|---|---|
| $\log ecex_{i,t}$ | 12,544 | 0.155 | 0.096 | 0.052 | 0.464 |
| $\log ecim_{i,t}$ | 12,544 | 0.098 | 0.370 | 0.047 | 0.342 |
| $\log pgdp_{i,t}$ | 12,544 | 2.196 | 3.172 | 4.745 | 14.053 |
| $\log pop_{i,t}$ | 12,544 | 0.540 | 0.702 | 0.336 | 1.235 |
| $\log rp_{i,t}$ | 12,544 | 2.397 | 5.001 | −10.694 | 22.266 |
| $\log il_{i,t}$ | 12,544 | 4.848 | 0.749 | 1.416 | 5.145 |
| $\log ofdi_{i,t-1}$ | 12,544 | 3.876 | 0.940 | 0.326 | 4.512 |

**Table 2.** OECD Countries List.

| No | Country | No | Country | No | Country | No | Country | No | Country |
|---|---|---|---|---|---|---|---|---|---|
| 1 | Australia | 7 | Estonia | 13 | Ireland | 19 | New Zealand | 25 | Sweden |
| 2 | Austria | 8 | Finland | 14 | Israel | 20 | The Netherlands | 26 | Switzerland |
| 3 | Belgium | 9 | France | 15 | Italy | 21 | Norway | 27 | UK |
| 4 | Canada | 10 | Germany | 16 | Japan | 22 | Poland | 28 | USA |
| 5 | Chile | 11 | Greece | 17 | Korea | 23 | Portugal | 29 | Luxembourg |
| 6 | Czech | 12 | Hungary | 18 | Latvia | 24 | Slovakia | 30 | Spain |
| 31 | Denmark | 32 | Iceland | | | | | | |

Table 1 presents the basic description of the variables used in this paper. The data include 12,544 observations for each variable. The mean of cross-border e-commerce export trade is 0.155, with a standard deviation equal to 0.096, a minimum value equal to 0.052 and a maximum value equal to 0.464. The mean of cross-border e-commerce import trade is 0.098 with a standard deviation equal to 0.370, a minimum value equal to 0.047, and a maximum value equal to 0.342. The mean of GDP per capita is 2.196 with a standard deviation equal to 3.172, a minimum value equal to 4.745, and a maximum value equal

to 14.053. The mean of the population is 0.540 with a standard deviation equal to 0.702, a minimum value equal to 0.336, and a maximum value equal to 1.235. The mean of the relative price level is 2.397 with a standard deviation equal to 5.001, a minimum value equal to −10.694, and a maximum value equal to 22.266. The mean of international logistics is 4.848 with a standard deviation equal to 0.749, a minimum value equal to 1.416, and a maximum value equal to 5.145. The mean of one-lagged period outward foreign direct investment is 3.876 with a standard deviation equal to 0.940, a minimum value equal to 0.326, and a maximum value equal to 4.512.

Table 2 presents the countries used in this paper. As we know, the Organization for Economic Co-operation and Development includes 38 countries. However, due to data unavailability, only 30 countries have been employed to test the propositions of this paper.

### 4.2. Panel Unit Root Test

In this paper, five kinds of panel unit root test approaches, including the IPS test, ADF test, PP test, LLC test, and HADRI test, will be employed to perform the panel unit root test, respectively. The results of these panel unit root tests are shown in Table 3.

**Table 3.** Results of Panel Unit Root Test.

| Approach | Statistics | Variable | | | | | | |
|---|---|---|---|---|---|---|---|---|
| | | $\log ecex_{i,t}$ | $\log ecim_{i,t}$ | $\log dgdp_{i,t}$ | $\log pop_{i,t}$ | $\log rp_{i,t}$ | $\log ofdi_{t-1}$ | $\log il_{i,t}$ |
| IPS | W-stat | −2.269 | −2.719 | −5.241 | −15.356 | −10.007 | −4.607 | −2.224 |
| | P-vale | 0.012 | 0.003 | 0.000 | 0.000 | 0.000 | 0.000 | 0.013 |
| ADF | $\chi^2$ | 59.927 | 81.740 | 190.186 | 168.503 | 182.945 | 126.236 | 89.284 |
| | P-vale | 0.621 | 0.067 | 0.000 | 0.000 | 0.000 | 0.000 | 0.020 |
| PP | $\chi^2$ | 65.685 | 77.170 | 163.278 | 54.741 | 298.385 | 151.444 | 67.258 |
| | P-vale | 0.418 | 0.125 | 0.000 | 0.789 | 0.000 | 0.000 | 0.366 |
| LLC | T-stat | −2.017 | −0.907 | −10.230 | −17.196 | −13.766 | −8.504 | −3.859 |
| | P-vale | 0.022 | 0.182 | 0.000 | 0.000 | 0.000 | 0.000 | 0.001 |
| HADRI | Z-stat | 7.989 | 9.506 | 6.549 | 9.095 | 10.199 | 9.757 | 7.494 |
| | P-vale | 0.000 | 0.000 | 0.000 | 0.000 | 0.000 | 0.000 | 0.000 |
| Approach | Statistics | $\Delta \log ecex_{i,t}$ | $\Delta \log ecim_{i,t}$ | $\Delta \log dgdp_{i,t}$ | $\Delta \log pop_{i,t}$ | $\Delta \log rp_{i,t}$ | $\Delta \log ofdi_{t-1}$ | $\Delta \log il_{i,t}$ |
| IPS | W-stat | −0.431 | −0.808 | −0.231 | −0.861 | −0.558 | −0.561 | −0.334 |
| | P-vale | 0.731 | 0.514 | 0.844 | 0.504 | 0.499 | 0.501 | 0.822 |
| ADF | $\chi^2$ | 228.038 | 218.479 | 247.742 | 300.641 | 254.665 | 322.513 | 176.136 |
| | P-vale | 0.000 | 0.000 | 0.000 | 0.000 | 0.000 | 0.000 | 0.000 |
| PP | $\chi^2$ | 314.154 | 364.121 | 488.073 | 109.429 | 525.135 | 481.495 | 234.103 |
| | P-vale | 0.000 | 0.000 | 0.000 | 0.000 | 0.000 | 0.000 | 0.000 |
| LLC | T-stat | −13.716 | −15.089 | −17.933 | −17.725 | −17.368 | −19.340 | −13.041 |
| | P-vale | 0.000 | 0.000 | 0.000 | 0.000 | 0.000 | 0.000 | 0.000 |
| HADRI | Z-stat | 0.708 | 0.506 | 0.819 | 0.159 | 0.136 | 0.329 | 0.579 |
| | P-vale | 0.514 | 0.709 | 0.438 | 0.911 | 0.925 | 0.832 | 0.715 |

Note: LLC indicates that the null hypothesis means that the unit root exists in the sequence. IPS indicates that the null hypothesis means that the unit root does not exist in the sequence. ADF indicates that the null hypothesis means that the unit root exists in the sequence. PP indicates that the null hypothesis means that the unit root exists in the sequence. HADRI indicates that the null hypothesis means that the unit root does not exist in the sequence. $\Delta$ indicates the first difference operator.

Table 3 shows the results of five kinds of unit root tests. As for the results of the IPS panel unit root test, the null hypothesis that the unit root does not exist is rejected. That is to say that all variables are non-stationary. However, taking the first difference, all of them turn stationary. As for the results of the ADF panel unit root test, the null hypothesis that the unit root exists is rejected, except for in the cross-border e-commerce export trade. After taking the first difference, it becomes stationary. As for the results of the PP panel unit root test, the null hypothesis that the unit root exists is rejected except

for in the cross-border e-commerce export trade, cross-border e-commerce import trade, population and international logistics. Taking the first difference, all of them turn stationary. As for the LLC panel unit root test, the null hypothesis that the unit root exists is rejected, except for in the cross-border e-commerce import trade. Taking the first difference, it turns stationary. As for the HADRI panel unit test, the null hypothesis that the unit root does not exist is rejected. This means that all variables are non-stationary. Taking the first difference, they turn stationary.

To summarise, the estimated results indicate that most of the variables are not stationary at levels. However, when these variables are taken as the first difference, all of them turn stationary. Therefore, it can be concluded that all the variables used in this paper are stationary after taking the first difference. As such, it is essential to perform a panel cointegration test among cross-border e-commerce export trade, cross-border e-commerce import trade, GDP per capita, population, relative price level, international logistics, and one-period lagged outward foreign direct investment.

*4.3. Panel Cointegration Test*

Based on the results of panel unit root tests and the model specification of the cointegration tests for long-run equilibrium in Section 3.3, the combined Johansen–Fisher cointegration test will be used to confirm the long-run relationship among these variables. The null hypothesis assumes that there is no cointegration among these variables. The combined Johansen–Fisher cointegration test will obey the vector auto-regressive process for the combination of the panel variables by utilizing the Fisher-Trace and Fisher-Maximum eigenvalue tests. The results will be shown in Tables 4–7.

**Table 4.** Results of Panel Cointegration Test.

| Trace Test | | | | Maximum Eigen Value Test | | | |
|---|---|---|---|---|---|---|---|
| $H_0$ | $H_1$ | $\lambda$-Trace Statistic | *p*-Value | $H_0$ | $H_1$ | $\lambda$-Max Statistic | *p*-Value |
| $t = 0$ | $t \geq 1$ | 273.9 | 0.000 *** | $t = 0$ | $t \geq 1$ | 273.9 | 0.000 *** |
| $t \leq 1$ | $t \geq 2$ | 430.6 | 0.000 *** | $t \leq 1$ | $t \geq 2$ | 314.8 | 0.000 *** |
| $t \leq 2$ | $t \geq 3$ | 849.0 | 0.000 *** | $t \leq 2$ | $t \geq 3$ | 587.7 | 0.000 *** |
| $t \leq 3$ | $t \geq 4$ | 590.8 | 0.000 *** | $t \leq 3$ | $t \geq 4$ | 590.8 | 0.000 *** |
| $t \leq 4$ | $t \leq 5$ | 26.34 | 1.000 | $t \leq 4$ | $t \geq 5$ | 284.2 | 0.000 *** |
| $t \leq 5$ | $t \geq 6$ | 42.98 | 0.988 | $t \leq 5$ | $t \geq 6$ | 79.82 | 0.118 |

$$\log ecex_{i,t} = 0.025 \log il_{i,t} - 0.047 \log ofdi_{i,t-1} + 0.062 \log pop_{i,t} + 0.052 \log dgdp_{i,t} - 0.057 \log rp_{i,t}$$
$$........................(6.441)............(-0.967)..................(10.071).................(4.165)................(-0.085)$$

Note: *** indicates the rejection of the null hypothesis of no cointegration at least at the 1% level of significance. Probabilities are computed using asymptotic Chi-square distribution. ( ) indicates the t-statistics.

**Table 5.** Results of Panel Cointegration Test.

| Trace Test | | | | Maximum Eigen Value Test | | | |
|---|---|---|---|---|---|---|---|
| $H_0$ | $H_1$ | $\lambda$-Trace Statistic | *p*-Value | $H_0$ | $H_1$ | $\lambda$-Max Statistic | *p*-Value |
| $t = 0$ | $t \geq 1$ | 273.9 | 0.000* ** | $t = 0$ | $t \geq 1$ | 273.9 | 0.000 *** |
| $t \leq 1$ | $t \geq 2$ | 430.6 | 0.000 *** | $t \leq 1$ | $t \geq 2$ | 314.8 | 0.000 *** |
| $t \leq 2$ | $t \geq 3$ | 849.0 | 0.000 *** | $t \leq 2$ | $t \geq 3$ | 587.7 | 0.000 *** |
| $t \leq 3$ | $t \geq 4$ | 590.8 | 0.000 *** | $t \leq 3$ | $t \geq 4$ | 590.8 | 0.000 *** |
| $t \leq 4$ | $t \geq 5$ | 35.06 | 0.999 | $t \leq 4$ | $t \geq 5$ | 403.5 | 0.000 *** |
| $t \leq 5$ | $t \geq 6$ | 44.36 | 0.981 | $t \leq 5$ | $t \geq 6$ | 62.78 | 0.590 |

$$\log il_{i,t} = 2.681 \log ecex_{i,t} + 1.138 \log ofdi_{i,t-1} - 0.457 \log pop_{i,t} + 0.053 \log dgdp_{i,t} + 0.011 \log rp_{i,t}$$
$$...................(6.441)...............(61.521).....................(-6.831)..................(4.030)...................(1.504)$$

Note: *** indicates the rejection of the null hypothesis of no cointegration at least at the 1% level of significance. Probabilities are computed using asymptotic Chi-square distribution. ( ) indicates the t-statistics.

**Table 6.** Results of Panel Cointegration Test.

| | Trace Test | | | | Maximum Eigen Value Test | | |
|---|---|---|---|---|---|---|---|
| $H_0$ | $H_1$ | $\lambda$-Trace Statistic | *p*-Value | $H_0$ | $H_1$ | $\lambda$-Max Statistic | *p*-Value |
| $t = 0$ | $t \geq 1$ | 271.5 | 0.000 *** | $t = 0$ | $t \geq 1$ | 271.5 | 0.000 *** |
| $t \leq 1$ | $t \geq 2$ | 426.6 | 0.000 *** | $t \leq 1$ | $t \geq 2$ | 304.8 | 0.000 *** |
| $t \leq 2$ | $t \geq 3$ | 941.4 | 0.000 *** | $t \leq 2$ | $t \geq 3$ | 699.6 | 0.000 *** |
| $t \leq 3$ | $t \geq 4$ | 573.8 | 0.000 *** | $t \leq 3$ | $t \geq 4$ | 573.8 | 0.000 *** |
| $t \leq 4$ | $t \geq 5$ | 20.79 | 0.000 *** | $t \leq 4$ | $t \geq 5$ | 352.4 | 0.000 *** |
| $t \geq 5$ | $t \geq 6$ | 45.75 | 0.973 | $t \leq 5$ | $t \geq 6$ | 45.75 | 0.973 |

$$\log ecim_{i,t} = 0.017 \log il_{i,t} + 0.013 \log ofdi_{i,t-1} + 0.093 \log pop_{i,t} + 0.015 \log dgdp_{i,t} + 0.019 \log rp_{i,t}$$
........................(11.354)...............(0.664)...................(3.789)....................(0.297)....................(0.702)

Note: *** indicates the rejection of the null hypothesis of no cointegration at least at the 1% level of significance. Probabilities are computed using asymptotic Chi-square distribution. ( ) indicates the t-statistics.

**Table 7.** Results of Panel Cointegration Test.

| | Trace Test | | | | Maximum Eigen Value Test | | |
|---|---|---|---|---|---|---|---|
| $H_0$ | $H_1$ | $\lambda$-Trace Statistic | *p*-Value | $H_0$ | $H_1$ | $\lambda$-Max Statistic | *p*-Value |
| $t = 0$ | $t \geq 1$ | 271.5 | 0.000 *** | $t = 0$ | $t \geq 1$ | 271.5 | 0.000 *** |
| $t \leq 1$ | $t \geq 2$ | 426.6 | 0.000 *** | $t \leq 1$ | $t \geq 2$ | 304.8 | 0.000 *** |
| $t \leq 2$ | $t \geq 3$ | 941.4 | 0.000 *** | $t \leq 2$ | $t \geq 3$ | 699.6 | 0.000 *** |
| $t \leq 3$ | $t \geq 4$ | 539.7 | 0.000 *** | $t \leq 3$ | $t \geq 4$ | 539.7 | 0.000 *** |
| $t \leq 4$ | $t \geq 5$ | 27.73 | 1.000 | $t \leq 4$ | $t \geq 5$ | 267.2 | 0.000 *** |
| $t \leq 5$ | $t \geq 6$ | 44.36 | 0.981 | $t \leq 5$ | $t \geq 6$ | 62.78 | 0.590 |

$$\log il_{i,t} = 1.036 \log ecex_{i,t} + 0.975 \log ofdi_{i,t-1} - 0.352 \log pop_{i,t} + 0.057 \log dgdp_{i,t} + 0.072 \log rp_{i,t}$$
.................(11.354).................(39.318)...................(−5.958)..................(4.743)................(1.089)

Note: *** indicates the rejection of the null hypothesis of no cointegration at least at the 1% level of significance. Probabilities are computed using asymptotic Chi-square distribution. ( ) indicates the t-statistics.

Table 4 presents the results of the cointegration test for model (1). Both Trace test and the Maximum eigenvalue test verify that the null hypothesis that no cointegration exists is rejected. Said differently, the cointegration exists for model (1). Meanwhile, the results also indicate that international logistics are positively related to cross-border e-commerce export trade and are significant in the statistics. Specifically, a 1% increase in international logistics leads to a 0.025% increase in cross-border e-commerce export trade. Moreover, this result verifies Hypothesis 1. What's more, this result is also consistent with the findings of Refs. [35,36].

Table 5 presents the results of the cointegration test for model (2). Both the Trace test and the Maximum eigenvalue test verify that the null hypothesis that no cointegration exists is rejected. Said differently, the cointegration exists for model (2). At the same time, the results also indicate that the cross-border e-commerce export trade is positively related to international logistics and is significant in the statistics. In more concrete terms, a 1% increase in cross-border e-commerce export trade results in a 2.681% increase in international logistics. Furthermore, this result also verifies Hypothesis 1. Additionally, this result is in keeping with the results of Refs. [3,37].

Table 6 presents the results of the cointegration test for model (3). Both the Trace test and Maximum eigenvalue test verify that the null hypothesis that no cointegration exists is rejected. Said differently, the cointegration exists for model (3). Meanwhile, the results also indicate that international logistics has a positive effect on cross-border e-commerce import trade and is significant in the statistic. Concretely, a 1% increase in international logistics brings about a 0.017% increase in cross-border e-commerce import trade. In addition, this result also verifies Hypothesis 3. Moreover, this result is in accordance with the ideas of Refs. [38,39].

Table 7 presents the results of the cointegration test for model (4). Both the Trace test and the Maximum eigenvalue test verify that the null hypothesis that no cointegration exists is rejected. Said differently, the cointegration exists for model (4). Meanwhile, the results also indicate that the cross-border e-commerce import trade has a positive effect on international logistics and is significant in the statistic. Said differently, a 1% increase in cross-border e-commerce import trade generates a 1.036% increase in international logistics. In addition, this result also verifies Hypothesis 3. Moreover, this result is identical to the findings of Refs. [40–42].

Based on the results of Tables 4–7, it can be confirmed that the cointegrating vectors of cross-border e-commerce export trade, cross-border e-commerce import trade, GDP per capita, population, relative price, international logistics and one-period lagged outward foreign direct investment exhibit panel cointegration. That is to say, there is long-term equilibrium among these variables in the case of OECD countries.

### 4.4. Panel Causality Test

Even though the results of combined Johansen–Fisher cointegration tests reveal that a long-term relationship among these variables exists, it cannot be confirmed what the magnitude of the relationship is. A bivariate panel causality test, which is based on the Dumitrescu–Hurlin process, will be used to test the causality relationship among these variables. The results of the panel causality tests are shown in Table 8.

**Table 8.** Pairwise Dumitrescu–Hurlin Panel Causality Test.

| Model | Hull Hypothesis | W-Statistic | Zbar-Statistic | *p*-Value | Conclusion |
|---|---|---|---|---|---|
| 19 | $\log ecex$ does not homogeneously cause $\log il$ | 9.771 | 6.768 | 0.000 | Rejected |
| 20 | $\log il$ does not homogeneously cause $\log ecex$ | 5.934 | 2.268 | 0.023 | Rejected |
| 21 | $\log ecim$ does not homogeneously cause $\log il$ | 6.073 | 2.431 | 0.015 | Rejected |
| 22 | $\log il$ does not homogeneously cause $\log ecim$ | 7.035 | 3.559 | 0.000 | Rejected |

According to the results of Table 8, it can be seen that the null hypotheses from model (19) to model (22) are rejected at 5% significant levels. Moreover, with a sample of OECD countries, the stronger bidirectional causal relationship between cross-border e-commerce export and international logistics, and the relationship between cross-border e-commerce import and international logistics, can be verified. Stated differently, the feedback causal relationship between cross-border e-commerce export and international logistics, and the relationship between cross-border e-commerce import and international logistics, exist within a sample of OECD countries.

### 4.5. Panel Vector Error Correction Model

In this paper, the panel vector error correction model was used to explore the short-term relationship among cross-border e-commerce export trade, cross-border e-commerce import trade, GDP per capita, population, relative price level, international logistics, and one-period lagged outward foreign direct investment. The results of panel vector error correction models (PVECM) are shown in Table 9.

**Table 9.** Results of Panel Vector Error Correction Model (PVECM).

| Variable | Model (5) | Model (6) | Model (7) | Model (8) |
|---|---|---|---|---|
| | $\Delta \log ecex_{i,t}$ | $\Delta \log il_{i,t}$ | $\Delta \log ecim_{i,t}$ | $\Delta \log il_{i,t}$ |
| $\lambda_i - ecm_{i,t-1}$ | −0.236*** (−4.255) | −0.022 ** (−2.502) | −0.472 *** (−8.617) | −0.103 * (−1.896) |
| $\Delta \log il_{i,t}$ | −0.065 *** (−4.221) | | −0.011 ** (−2.477) | |
| $\Delta \log ecex_{i,t}$ | | −0.383 ** (−2.342) | | |
| $\Delta \log ecim_{i,t}$ | | | | −0.306 * (−1.734) |
| $\Delta \log dgdp_{i,t}$ | 0.052 * (1.806) | −0.032 (−1.476) | 0.021 (0.540) | −0.025 (−1.356) |
| $\Delta \log pop_{i,t}$ | 0.039 (0.265) | 0.039 (0.243) | 0.066 (0.013) | 0.120 (0.825) |
| $\Delta \log rp_{i,t}$ | 0.018 (0.986) | 0.080 ** (2.531) | −0.053 (−0.247) | 0.015 * (1.873) |
| $\Delta \log odfi_{i,t-1}$ | −0.038 ** (−2.142) | −0.012 (−0.136) | −0.036 ** (−2.447) | 0.013 * (1.878) |
| Category | Cross-border e-commerce export trade and international logistic | | Cross-border e-commerce import trade and international logistic | |

Note: ( ) indicates the t-statistics. * indicates 10% significance level. ** indicates 5% significance level. *** indicates the 1% significance level.

Table 9 shows the results of the panel vector error correction estimation for model (5), model (6), model (7), and model (8), respectively. For model (5), $\lambda_1$ is equal to −0.236 and is significant at the 1% level. Moreover, this result also verifies that the panel cointegration relationship exists. Namely, model (1) holds. This suggests that deviation from the cointegration system of cross-border e-commerce export trade will lead to the cross-border e-commerce export trade changing by approximately 23.6% in the next period. Meanwhile, it can be also found that international logistics has a negative effect on cross-border e-commerce export trade. This result verifies Hypothesis 2. For model (6), $\lambda_2$ is equal to −0.022 and significant at the 5% level. This result also implies that the panel cointegration relationship exists. In other words, model (2) holds. This suggests that deviation from the cointegration system of international logistics will lead to the international logistics changing by approximately 2.200% in the next period. At the same time, the cross-border e-commerce export trade has a negative effect on international logistics. This result verifies Hypothesis 4. Compared with the estimating value of $\lambda_1$ and $\lambda_2$, the ability to return to the long-term equilibrium of deviation from the cointegration system of cross-border e-commerce export trade in the short term is stronger than that for the deviation from the cointegration system of international logistics. For model (7), $\lambda_3$ is equal to −0.472 and significant at the 1% level. Moreover, this result also verifies that the panel cointegration relationship exists. Said differently, model (3) holds. This indicates that the deviation from the cointegration system of cross-border e-commerce import trade will lead to the cross-border e-commerce import trade changing approximately 47.2% in the next period. Simultaneously, international logistics has a negative effect on cross-border e-commerce import trade. In fact, Niu [43] agreed with this finding. Moreover, this result verifies Hypothesis 2. For model (8), $\lambda_4$ is equal to −0.103 and significant at the 10% level. Furthermore, these results also verify that the panel cointegration relationship exists. Stated differently, model (4) holds. This shows that the deviation from the cointegration system of international logistics will lead to the international logistics changing approximately 10.3% in the next period. Meanwhile, cross-border e-commerce import trade has a negative effect on international logistics. As a matter of fact, this result responds well to the

idea of Ref. [44]. Furthermore, this result verifies Hypothesis 4. When taking the estimating value of $\lambda_3$ and $\lambda_4$ into consideration, the ability to return to the long-term equilibrium of the deviation from the cointegration system of cross-border e-commerce import trade in the short term is stronger than that for the deviation from the cointegration system of international logistics.

In addition, the results of the panel vector error correction model have vital economic and trade policy implications. An increase in the GDP per capita will achieve a breakthrough in the cross-border e-commerce export trade. An increased relative price level will restrain the development of international logistics in terms of cross-border e-commerce import trade. More importantly, the one-period lagged outward foreign direct investment has a crowding-out effect on cross-border e-commerce trade. However, it has a crowding-in effect on international logistics in terms of cross-border e-commerce import trade.

## 5. Conclusions and Suggestions

Economic globalization provides a new impetus for the development of cross-border e-commerce trade and international logistics. As the history of the development of cross-border e-commerce trade is relatively short, the data collection is very difficult. Due to this limitation, there are few empirical studies on the relationship between cross-border e-commerce trade and international logistics. Even so, the current research reaches no consensus about this proposition. Due to this background, this paper regards OECD countries as a sample to explore the dynamic relationship among cross-border e-commerce export trade, cross-border e-commerce import trade, GDP per capita, population, relative price level, international logistics and one-period lagged outward foreign direct investment. The panel data from 2000 to 2018 will be employed to fulfil an empirical analysis under a series of econometric approaches, such as panel unit root tests, panel cointegration tests, panel causality tests, and the panel vector error correction model. The results of the panel unit root test illustrate that most variables are generated by the panel unit root process. The results of the combined Johansen–Fisher cointegration test indicate that there is a long-run relationship among these variables. The results of the panel causality test confirm the existence of the stronger bidirectional causal relationship between cross-border e-commerce export and international logistics, and the relationship between cross-border e-commerce import and international logistics. Finally, the results of the panel vector error correction model reveal the short-term relationship between these variables.

In the long run, the interaction between international logistics and cross-border e-commerce trade is positive and significant in terms of statistics. In terms of cross-border e-commerce export trade, the population and the GDP per capita have a positive effect on cross-border e-commerce export trade. However, the one-period lagged outward foreign direct investment has a negative effect on cross-border e-commerce export trade. Meanwhile, the one-period lagged outward foreign direct investment and the GDP per capita have a positive effect on international logistics. However, the population has a negative effect on international logistics. In terms of cross-border e-commerce import trade, the population has a positive effect on cross-border e-commerce import trade. The one-period lagged outward foreign direct investment and the GDP per capita have a positive effect on international logistics. However, the population has a negative effect on international logistics.

In the short run, the interaction between international logistics and cross-border e-commerce trade is negative and significant in terms of statistics. In terms of cross-border e-commerce export trade, the GDP per capita has a positive effect on cross-border e-commerce export trade. The one-period lagged outward foreign direct investment has a negative effect on cross-border e-commerce export trade. The relative price has a positive effect on international logistics. In terms of cross-border e-commerce import trade, one-period lagged outward foreign direct investment has a negative effect on cross-border e-commerce import trade. The relative price and one-period lagged outward foreign direct investment have a positive effect on international logistics. Meanwhile, this suggests that

deviation from the cointegration system of cross-border e-commerce trade and international logistics will lead the cross-border e-commerce trade and international logistics to change within a range of approximately 2.2% to 47.2% in the next period.

According to the empirical evidence this paper provides, some corresponding suggestions will be put forward. For example, the GDP per capita is an important factor affecting the dynamic relationship between international logistics and cross-border e-commerce trade. Therefore, OECD countries should expand the production to increase the GDP per capita. When taking the long-term dynamic relationship among them, the dynamic relationship between international logistics and cross-border e-commerce trade is positive. Therefore, OECD countries should take up some related policies, such as reductions of the tariff and improvements of the logistics infrastructure, in order to promote the sustainable development of international logistics. Then, an increase in the international logistics can result in an increase in the cross-border e-commerce trade. When taking the short-run dynamic relationship among them, the relative price level also plays a vital role in affecting the short-term dynamic relationship between international logistics and cross-border e-commerce trade. OECD countries should take up some related policies, such as money demand and money supply, to control the price level. The reason is that the appropriate price level can be beneficial for the sustainable development of international logistics and cross-border e-commerce trade.

To this end, there are some limitations in this paper. The heterogeneity among selected OECD countries is ignored. Therefore, one possible extension for future research in this area may be to segment the full sample into some sub-samples to confirm the evolution of the relationship among them. Furthermore, another possible extension of this investigation may include a proxy of international logistics (in this paper, it is defined as the international freight transport in million tons per kilometer). For cross-border e-commerce trade, this paper only employs the total volume of cross-border e-commerce export trade and the total volume of cross-border e-commerce import trade. In future works, if possible, the volume of cross-border e-commerce trade can be subdivided into B2B, B2C, O2O and G2B. This settlement seems to be more interesting in terms of producing good works. Due to data unavailability, the size of the research sample is extremely limited. Therefore, a possible way to ensure the estimated results are reliable and robust is to expand the size of the research sample and to lengthen the time span.

**Author Contributions:** Conceptualization, Y.H. and Y.-J.C.; methodology, Y.H. and Y.-J.C.; software, Y.H.; validation, Y.H. and R.W.; formal analysis, R.W. and Y.-J.C.; investigation, R.W.; resources, Y.H.; data curation, R.W.; writing—original draft preparation, Y.H.; writing—review and editing, R.W.; visualization, Y.H. and Y.-J.C.; supervision, Y.H. and Y.-J.C.; project administration, Y.H and Y.-J.C. All authors have read and agreed to the published version of the manuscript.

**Funding:** This research received no external funding.

**Informed Consent Statement:** Not applicable.

**Data Availability Statement:** The data presented in this study are available on the World Bank (https://data.worldbank.org/), the Data Center of Organization for Economic Co-operation and Development (https://data.oecd.org/), the United Nations Conference on Trade and Development (https://unctad.org/), Statista (https://www.statista.com/) and iimedia (https://data.iimedia.cn/).

**Conflicts of Interest:** The authors declare no conflict of interest.

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
