# Peer review of "International Logistics and Cross-Border E-Commerce Trade: Who Matters Whom?"

_sustainability, doi:10.3390/su13041745_

Round 1
Reviewer 1 Report
The article needs more specified explanation for both the goal of the research and research hypothesis that enabled the realization of author’s assumptions. Moreover, the presented theoretical approach and coverage of literature need completion as well. The introduction needs to be improved in order to make the research question clear, as well as the contribution of the work to the existing literature.
Furthermore, it is advisable to present information on current findings regarding the issues in question and juxtapose conclusions drawn by the authors with them. Such a comparative analysis of the obtained results juxtaposed with the earlier findings presented by other authors would enrich the article and would enable a broader discussion regarding the level of international trade in transport services. The cause and effect relationships of the analyzed phenomena were not indicated either. It is worth pointing to the limitations of research and directions of future analyzes
Author Response
- The article needs more specified explanation for the goal of the research.
Reply:
Unlike previous papers in the international logistics - cross-border e-commerce trade, this paper sets Organization for Economic Co-operation and Development countries as an example to explore the dynamic interaction between international logistics and cross-border e-commerce trade.
- research hypothesis that enabled the realization of author’s assumptions.
Reply:
Hypothesis One: International logistics is positively related to the sustainable development of cross-border e-commerce trade in the long run.
Hypothesis Two: International logistics is negatively related to the sustainable development of cross-border e-commerce trade in the short run.
Hypothesis Three: Cross-border e-commerce trade is positively related to the sustainable development of international logistics in the long run.
Hypothesis Four: Cross-border e-commerce trade is negatively related to the sustainable development of international logistics in the short run.
All these hypotheses have been verified in the following empirical analyses.
- The introduction needs to be improved in order to make the research question clear.
Reply:
In reality, the sustainable development of both international logistics and cross-border e-commerce trade also faces many challenges. From the theoretical level, the rapid development of cross-border e-commerce trade can increase the transaction orders, which can promote the sustainable development of international logistics. However, in the actual cooperation, the coordination and sustainability of both international logistics and cross-border e-commerce trade are fragile. As we know, the e-commerce platform uses the Internet to establish the communication links with consumers, which provides consumers with more convenient product purchase channels. Transactions can be achieved only with the help of the network, which is not limited by both space and time. As a platform transportation partner, the international logistics needs to do a good job in customs such as quality inspection and other work after receiving the international logistics transportation requirements sent by the platform. As a matter of fact, each link is very complex and takes a lot of time. To solve the time-consuming problem, some cross-border e-commerce trade service providers choose to establish overseas storage model to offer customers with return and exchange services. Due to the large cost of overseas warehousing construction, the demand for product sales data information is high. At present, the international logistics enterprises can not obtain the product warehouse management data information, which leads to the low accuracy of warehouse design. The layout of storage space can not meet the actual storage needs, which is not conducive to the improvement of warehouse management level. And the ability of international logistics enterprises has always been difficult to maintain the sustainable development. However, the cross-border e-commerce platform does not know enough about overseas warehousing, and does not understand the importance of international logistics enterprises to their own sustainable development. As a result, the sustainability of bilateral cooperation is weak as a whole.
Furthermore, to improve the service, the cross-border e-commerce enterprises need to cooperate with international logistics enterprises to provide high-quality services for consumers. This operation goal needs to be completed by both parties’ synchronous operation. But in the actual operation process, the cross-border e-commerce enterprises focus on their own work. To let consumers experience high-quality shopping services, they do not consider whether the work of international logistics enterprises is difficult or not. In the international logistics transportation services, the cross-border e-commerce enterprises will set the cooperation requirements of the nearest warehouse delivery, which limits the sustainable operation of international logistics enterprises. Considering the feasibility of international logistics delivery, human resources, transportation costs and other issues, international logistics enterprises are unable to meet the requirements of this international distribution, which results in contradictions in the operation and cooperation between both two parties.
In addition, there are still many problems worth being discussed between cross-border e-commerce trade and international logistics. Therefore, a larger number of scholars related in this field have had a attempt to study this proposition. Ying [1] performs a research about logistics mode selection in B2C cross-border e-commerce in China. His findings show that the lagged construction of international logistics system and the high cost of international logistics have seriously hindered the further development of China's B2C cross-border e-commerce enterprises. Even though with different samples like India, Alyoubi [2] finds that international logistics problems pose as a significant barrier for the sustainable development of e-commerce trade. However, Wang and Lee [3] sets China as a sample to discuss the effect of international logistics on international trade. They find that this kind of effect is positive. Meanwhile, Cho, et al [4] also find that the international logistics capability is positively correlated with the enterprises performance in terms of cross-border e-commerce market. Up to this day, although a great many of relating experts have deeply explored the dynamic relationship between cross-border e-commerce trade and international logistics, they have not reached a consensus. As a matter of fact, due to different perspectives, different samples, different times spans and different research approaches, drawing different conclusion about this proposition can be understood. Said differently, it is fully valuable to exploit the relationship between cross-border e-commerce trade and international logistics. because of this background, this paper sets Organization for Economic Co-operation and Development (OECD) countries as an example over the period 2000-2018 to explore the dynamic relationship between international logistics and cross-border e-commerce trade.
- The contribution of the work to the existing literature.
Reply:
The contribution of this current paper to the previous empirical literature focusing on the relationship between cross-border e-commerce trade and international logistics is four fold. First, this paper attempts to fill the gap in the empirical literature with regard to the dynamic relationship between cross-border e-commerce trade and international logistics in the Organization for Economic Co-operation and Development (OECD) countries. The examination of the Organization for Economic Co-operation and Development (OECD) countries as a case study is particularly interesting since most of the Organization for Economic Co-operation and Development (OECD) countries are open enough to each other and fully integrated in economics, they have a high potential for performing economic activities from new trade modes and new logistics modes. Meanwhile, most of them have a huge capacity of producing a great quantity of goods and platforms to conduct the cross-border e-commerce trade. Moreover, most of them also have extremely advanced international logistics system to satisfy the needs of cross-border e-commerce trade activities. These basic fundamentals enable our results more reliable and robust. Second, unlike previous researches that have used the traditional econometric models such as vector auto-regression model and vector error correction model, this paper employs a recent multivariate economic technique which is famous as the panel data vector error correction model. Because the properties of both time series and cross-section of data are taken into consideration, this can help us more accurately to estimate the long-run and short-run relationship between cross-border e-commerce trade and international logistics. Third, a large number of panel unit root tests are used to test stationarity of variables used in this paper. This can also ensure the accuracy of empirical results. Finally, these evidences this paper provided can form a foundation for the future scholars who are interested in exploring the dynamic relationship between cross-border e-commerce trade and international logistics.
- Furthermore, it is advisable to present information on current findings regarding the issues in question and juxtapose conclusions drawn by the authors with them. Such a comparative analysis of the obtained results juxtaposed with the earlier findings presented by other authors would enrich the article and would enable a broader discussion regarding the level of international trade in transport services.
Reply:
I have done this part according to your valuable advice. The revised content will be presented in body of this paper.
- The cause and effect relationships of the analyzed phenomena were not indicated either.
Reply:
Table 8. Pairwise Dumitrescu-Hurlin Panel Causality Test
|
Model |
Hull Hypotesis |
W-statistic |
Zbar-statistic |
P-value |
Conclusion |
|
19 |
does not homogeneously cause |
9.771 |
6.768 |
0.000 |
Rejected |
|
20 |
does not homogeneously cause |
5.934 |
2.268 |
0.023 |
Rejected |
|
21 |
does not homogeneously cause |
6.073 |
2.431 |
0.015 |
Rejected |
According to the results of <Table 8>, it can be seen that the null hypotheses from model (19) to model (22) are rejected at 5% significant levels. Moreover, with a sample of OECD countries, the stronger bidirectional causal relationship between cross-border e-commerce export and international logistics, and the relationship between cross-border e-commerce import and international logistics can be verified. Stated differently, the feedback causal relationship between cross-border e-commerce export and international logistics, and relationship between cross-border e-commerce import and international logistics exist with a sample of OECD countries.
- It is worth pointing to the limitations of research and directions of future analyzes.
Reply:
To this end, there are some limitations in this paper. the heterigeneity among selected OECD countries is ignored. Therefore, one possible extension for future researches of this attempt may be to segment the full sample into some sub-samples to confirm the evolution of the relationship among them. Furthermore, another possible extension of this investigation may include proxy of international logistics (in this paper, it is defined as the international freight transport million tone per kilometer). For cross-border e-commerce trade, this paper only employs the total volume of cross-border e-commerce export trade and total volume of cross-border e-commerce import trade. In the future works, if possible, the volume of cross-border e-commerce trade can be subdivided into B2B, B2C, O2O and G2B. This settlement seems to be more interesting to produce good works. Due to data unavailability, the size of research sample is extremely limited. Therefore, a possible way to let the estimating results be reliable and robust is to expand the size of research sample and to lengthen the time span.

Reviewer 2 Report
I think that the paper should include a bit more debate on the existing literature, since the one that has been reflected seems scarce.
Authors can include and coment references such as:
Gomez-Herrera, E., Martens, B., & Turlea, G. (2014). The drivers and impediments for cross-border e-commerce in the EU. Information Economics and Policy, 28, 83-96.
Thai Young Kim, Rommert Dekker & Christiaan Heij (2017) Cross-Border Electronic Commerce: Distance Effects and Express Delivery in European Union Markets, International Journal of Electronic Commerce, 21:2, 184-218.
Giuffrida, M., Mangiaracina, R., Perego, A. and Tumino, A. (2017), "Cross-border B2C e-commerce to Greater China and the role of logistics: a literature review", International Journal of Physical Distribution & Logistics Management, Vol. 47 No. 9, pp. 772-795.
Wang, Yu; Wang, Yi; Lee, Soo H. 2017. "The Effect of Cross-Border E-Commerce on China’s International Trade: An Empirical Study Based on Transaction Cost Analysis" Sustainability 9, no. 11: 2028.
Ma, S., Chai, Y., & Zhang, H. (2018). Rise of Cross‐border E‐commerce Exports in China. China & World Economy, 26(3), 63-87.
Cui, Y., Mou, J., Cohen, J., & Liu, Y. (2019). Understanding information system success model and valence framework in sellers’ acceptance of cross-border e-commerce: a sequential multi-method approach. Electronic Commerce Research, 19(4), 885-914.
Mou, J., Ren, G., Qin, C., & Kurcz, K. (2019). Understanding the topics of export cross-border e-commerce consumers feedback: an LDA approach. Electronic Commerce Research, 19(4), 749-777.
Author Response
- I think that the paper should include a bit more debate on the existing literature, since the one that has been reflected seems scarce.Authors can include and comment references such as:
Reply:
I have discussed more previous researches and those researches including your suggesting literature. All will be presented as follows:
Ahi, P., & Searcy, C. (2013). A comparative literature analysis of definitions for green and sustainable supply chain management. Journal of cleaner production, 52, 329-341.
Alyoubi, A. A. (2015). E-commerce in developing countries and how to develop them during the introduction of modern systems. Procedia Computer Science, 65, 479-483.
Cho, J. J. K., Ozment, J., & Sink, H. (2008). Logistics capability, logistics outsourcing and firm performance in an e‐commerce market. International journal of physical distribution & logistics management.
Cui, Y., Mou, J., Cohen, J., & Liu, Y. (2019). Understanding information system success model and valence framework in sellers’ acceptance of cross-border e-commerce: a sequential multi-method approach. Electronic Commerce Research, 19(4), 885-914.
Delfmann, W., Albers, S., & Gehring, M. (2002). The impact of electronic commerce on logistics service providers. International journal of physical distribution & logistics management.
Giuffrida, M., Mangiaracina, R., Perego, A., & Tumino, A. (2017). Cross-border B2C e-commerce to Greater China and the role of logistics: a literature review. International Journal of Physical Distribution & Logistics Management.
Gomez-Herrera, E., Martens, B., & Turlea, G. (2014). The drivers and impediments for cross-border e-commerce in the EU. Information Economics and Policy, 28, 83-96.
Gong, R., Xue, J., Zhao, L., Zolotova, O., Ji, X., & Xu, Y. (2019). A bibliometric analysis of green supply chain management based on the Web of Science (WOS) platform. Sustainability, 11(12), 3459.
Hesse, M. (2002). Shipping news: the implications of electronic commerce for logistics and freight transport. Resources, conservation and recycling, 36(3), 211-240.
Hong, J. Y., Han, H. N., & Shim, C. Y. (2018). a study of the vitalization of cross-border e-commerce between ASEAN and Korea: Focus on trade and logistics issue. Journal of International Trade & Commerce, 14(2), 179-195.
Ji, S., & Sun, Q. (2017). Low-carbon planning and design in B&R logistics service: a case study of an e-commerce big data platform in China. Sustainability, 9(11), 2052.
Kadłubek, M. (2015). The Selected Areas of E-logistics in Polish E-commerce. Procedia Computer Science, 65, 1059-1065.
Kim, T. Y., Dekker, R., & Heij, C. (2017). Cross-border electronic commerce: Distance effects and express delivery in European Union markets. International Journal of Electronic Commerce, 21(2), 184-218.
Kusi-Sarpong, S., Gupta, H., & Sarkis, J. (2019). A supply chain sustainability innovation framework and evaluation methodology. International Journal of Production Research, 57(7), 1990-2008.
Leinbach, T. R. (2007). Globalized freight transport: intermodality, e-commerce, logistics and sustainability. Edward Elgar Publishing.
Ma, S., Chai, Y., & Zhang, H. (2018). Rise of Cross‐border E‐commerce Exports in China. China & World Economy, 26(3), 63-87.
Mou, J., Ren, G., Qin, C., & Kurcz, K. (2019). Understanding the topics of export cross-border e-commerce consumers feedback: an LDA approach. Electronic Commerce Research, 19(4), 749-777.
Niu, B., Wang, J., Lee, C. K., & Chen, L. (2019). “Product+ logistics” bundling sale and co-delivery in cross-border e-commerce. Electronic Commerce Research, 19(4), 915-941.
Pourhejazy, P. (2020). Destruction Decisions for Managing Excess Inventory in E-Commerce Logistics. Sustainability, 12(20), 8365.
Qin, X., Liu, Z., & Tian, L. (2020). The optimal combination between selling mode and logistics service strategy in an e-commerce market. European Journal of Operational Research, 289(2), 639-651.
Schöder, D. (2016). The impact of e-commerce development on urban logistics sustainability. Open Journal of Social Sciences, 4(03), 1.
Sroufe, R., & Joseph, S. (Eds.). (2017). Strategic sustainability: the state of the art in corporate environmental management systems.
Uygun, Ö., & Dede, A. (2016). Performance evaluation of green supply chain management using integrated fuzzy multi-criteria decision making techniques. Computers & Industrial Engineering, 102, 502-511.
Wang, T., Kang, J. W., & Valentine, V. F. (2020). A holistic analysis of national e-commerce and logistics development. Maritime Economics & Logistics, 1-14.
Wang, Y., & Lee, S. H. (2017). The effect of cross-border e-commerce on China’s international trade: An empirical study based on transaction cost analysis. Sustainability, 9(11), 2028.
Zimon, D., Tyan, J., & Sroufe, R. (2019). Implementing sustainable supply chain management: reactive, cooperative, and dynamic models. Sustainability, 11(24), 7227.
Zimon, D., Tyan, J., & Sroufe, R. (2020). Drivers of sustainable supply chain management: Practices to alignment with un sustainable development goals. International Journal for Quality Research, 14(1).
Żurek, J. (2015). E-commerce influence on changes in logistics processes. LogForum, 11(2), 129-138.

Reviewer 3 Report
It is a good manuscript with potential for publication. But some comments and some improvements are suggested below.
1. It is not clear how your research contributes to sustainable development. How sustainability is maintained by your research is not define anywhere. Therefore, you need to emphasize the link between the article and the journal's profile in the introduction, literature review and conclusions.
2. Introduction and Literature review section are fall to provide exact novelty, finding, and problem definition. Thus, authors are advised to recreate those section very carefully. In my opinion, the literature review is practically non-existent.Literature review should be keyword basis. Break this section down into a subsection and expand. I suggest starting with the section on SSCM. Indicate how your research supports SSCM.
It is worth to refer to the works of the following authors:
- Gong, R.; Xue, J.; Zhao, L.; Zolotova, O.; Ji, X.; Xu, Y. A Bibliometric Analysis of Green Supply Chain Management Based on the Web of Science (WOS) Platform. Sustainability 2019, 11, 3459.
- Zimon, D., Tyan, J., & Sroufe, R. (2020). DRIVERS OF SUSTAINABLE SUPPLY CHAIN MANAGEMENT: PRACTICES TO ALIGNMENT WITH UN SUSTAINABLE DEVELOPMENT GOALS. International Journal for Quality Research, 14(1).
- Zimon, D.; Tyan, J.; Sroufe, R. Implementing Sustainable Supply Chain Management: Reactive, Cooperative, and Dynamic Models. Sustainability 2019, 11, 7227.
- Sroufe, R.; Joseph, S. (Eds.)Strategic sustainability: The State of the Art in Corporate Environmental ManagementSystems; Routledge: Abingdon, UK, 2017
- Kusi-Sarpong, S.; Gupta, H.; Sarkis, J. A supply chain sustainability innovation framework and evaluationmethodology.Int. J. Prod. Res.2019,57, 1–19
- Ahi, P.; Searcy, C. A comparative literature analysis of definitions for green and sustainable supply chainmanagement.J. Clean. Prod.2013,52, 329–341.
- Uygun, Ö.; Dede, A. Performance evaluation of green supply chain management using integrated fuzzymulti-criteria decision making techniques.Comput. Ind. Eng.2016,102, 502–511
3. Need proper explanation of each table, such that reader can understand your work. Work especially on Tables 1-7.
4. In the Discussion section, the obtained results should be confronted with the opinions contained in the literature on the subject and with the results of the other studies, so that it is clear whether the obtained results confirm or question the current state of knowledge.
5. The basic pillars of the sustainability were not maintained by your research. Correct it.
Summarizing the article should be refined in the following areas: • Expand the theoretical part and enrich the research part with discussion, • Develop conclusions; The description of the methodology used and the different steps taken may be improved.
Good Luck !
Author Response
- It is not clear how your research contributes to sustainable development. How sustainability is maintained by your research is not define anywhere. Therefore, you need to emphasize the link between the article and the journal's profile in the introduction, literature review and conclusions.
Reply:
The cross-border e-commerce trade brings market for the development of international logistics. Traditional business models are increasingly unable to meet the needs of people. In the new era, consumers will pay more attention to both quality and variety of goods. In addition, consumers will also focus on the shopping experience. Fortunately, the emergence of cross-border e-commerce trade will greatly enhance the shopping convenience, meet the consumer demand and optimize the consumer shopping experience. Meanwhile, the emergence of cross-border e-commerce trade also plays a role in improving the quality of service, improving the effectiveness of supply chain, enhancing the efficiency of business operations and increasing the volume of international trade. Therefore, many traditional enterprises have introduced the models of cross-border e-commerce trade. The huge cross-border e-commerce trade market provides market opportunities for the development of international logistics and a necessary link for cross-border e-commerce trade. On the contrary, international logistics is an essential part of building a cross-border e-commerce trade supply chain. The process of cross-border e-commerce trade includes negotiation, contracting, payment, logistics and other aspects. The development of cross-border e-commerce trade also provides market opportunities for the development of enterprises related to these links. In the cross-border e-commerce trade, the basis of corporate and consumer contract practice is non-virtual “international logistics”. The factors affecting consumer consumption experience are also the efficiency and cost of logistics. Therefore, the cross-border e-commerce trade not only provides market opportunities for the development of international logistics, but also challenges its development. So, the international logistics is an indispensable link in the development of cross-border e-commerce trade. Its development level has also become a key factor in the integration of cross-border e-commerce trade supply chain.
In reality, the sustainable development of both international logistics and cross-border e-commerce trade also faces many challenges. From the theoretical level, the rapid development of cross-border e-commerce trade can increase the transaction orders, which can promote the sustainable development of international logistics. However, in the actual cooperation, the coordination and sustainability of both international logistics and cross-border e-commerce trade are fragile. As we know, the e-commerce platform uses the Internet to establish the communication links with consumers, which provides consumers with more convenient product purchase channels. Transactions can be achieved only with the help of the network, which is not limited by both space and time. As a platform transportation partner, the international logistics needs to do a good job in customs such as quality inspection and other work after receiving the international logistics transportation requirements sent by the platform. As a matter of fact, each link is very complex and takes a lot of time. To solve the time-consuming problem, some cross-border e-commerce trade service providers choose to establish overseas storage model to offer customers with return and exchange services. Due to the large cost of overseas warehousing construction, the demand for product sales data information is high. At present, the international logistics enterprises can not obtain the product warehouse management data information, which leads to the low accuracy of warehouse design. The layout of storage space can not meet the actual storage needs, which is not conducive to the improvement of warehouse management level. And the ability of international logistics enterprises has always been difficult to maintain the sustainable development. However, the cross-border e-commerce platform does not know enough about overseas warehousing, and does not understand the importance of international logistics enterprises to their own sustainable development. As a result, the sustainability of bilateral cooperation is weak as a whole.
Furthermore, to improve the service, the cross-border e-commerce enterprises need to cooperate with international logistics enterprises to provide high-quality services for consumers. This operation goal needs to be completed by both parties’ synchronous operation. But in the actual operation process, the cross-border e-commerce enterprises focus on their own work. To let consumers experience high-quality shopping services, they do not consider whether the work of international logistics enterprises is difficult or not. In the international logistics transportation services, the cross-border e-commerce enterprises will set the cooperation requirements of the nearest warehouse delivery, which limits the sustainable operation of international logistics enterprises. Considering the feasibility of international logistics delivery, human resources, transportation costs and other issues, international logistics enterprises are unable to meet the requirements of this international distribution, which results in contradictions in the operation and cooperation between both two parties.
In addition, there are still many problems worth being discussed between cross-border e-commerce trade and international logistics. Therefore, a larger number of scholars related in this field have had a attempt to study this proposition. Ying [1] performs a research about logistics mode selection in B2C cross-border e-commerce in China. His findings show that the lagged construction of international logistics system and the high cost of international logistics have seriously hindered the further development of China's B2C cross-border e-commerce enterprises. Even though with different samples like India, Alyoubi [2] finds that international logistics problems pose as a significant barrier for the sustainable development of e-commerce trade. However, Wang and Lee [3] sets China as a sample to discuss the effect of international logistics on international trade. They find that this kind of effect is positive. Meanwhile, Cho, et al [4] also find that the international logistics capability is positively correlated with the enterprises performance in terms of cross-border e-commerce market. Up to this day, although a great many of relating experts have deeply explored the dynamic relationship between cross-border e-commerce trade and international logistics, they have not reached a consensus. As a matter of fact, due to different perspectives, different samples, different times spans and different research approaches, drawing different conclusion about this proposition can be understood. Said differently, it is fully valuable to exploit the relationship between cross-border e-commerce trade and international logistics. because of this background, this paper sets Organization for Economic Co-operation and Development (OECD) countries as an example over the period 2000-2018 to explore the dynamic relationship between international logistics and cross-border e-commerce trade. Then, the panel data will be used to fulfil an empirical analysis under some econometric approaches such as panel unit root test, panel cointegration test, panel causality test and panel vector error correction model. Moreover, there are seven variables used in this paper. They are the cross-border e-commerce export trade, the cross-border e-commerce import trade, the GDP per capita, the population, the relative price, the international logistics and the one-period lagged outward foreign direct investment. Finally, all these variables and econometric methods will be employed to discuss the dynamic relationship between cross-border e-commerce trade and international logistics.
What’s more, I have modified another parts linked to sustainability in the body of this paper.
- Introduction and Literature review section are fall to provide exact novelty, finding, and problem definition. Thus, authors are advised to recreate those section very carefully. In my opinion, the literature review is practically non-existent.Literature review should be keyword basis. Break this section down into a subsection and expand. I suggest starting with the section on SSCM. Indicate how your research supports SSCM.
Reply:
In reality, the sustainable development of both international logistics and cross-border e-commerce trade also faces many challenges. From the theoretical level, the rapid development of cross-border e-commerce trade can increase the transaction orders, which can promote the sustainable development of international logistics. However, in the actual cooperation, the coordination and sustainability of both international logistics and cross-border e-commerce trade are fragile. As we know, the e-commerce platform uses the Internet to establish the communication links with consumers, which provides consumers with more convenient product purchase channels. Transactions can be achieved only with the help of the network, which is not limited by both space and time. As a platform transportation partner, the international logistics needs to do a good job in customs such as quality inspection and other work after receiving the international logistics transportation requirements sent by the platform. As a matter of fact, each link is very complex and takes a lot of time. To solve the time-consuming problem, some cross-border e-commerce trade service providers choose to establish overseas storage model to offer customers with return and exchange services. Due to the large cost of overseas warehousing construction, the demand for product sales data information is high. At present, the international logistics enterprises can not obtain the product warehouse management data information, which leads to the low accuracy of warehouse design. The layout of storage space can not meet the actual storage needs, which is not conducive to the improvement of warehouse management level. And the ability of international logistics enterprises has always been difficult to maintain the sustainable development. However, the cross-border e-commerce platform does not know enough about overseas warehousing, and does not understand the importance of international logistics enterprises to their own sustainable development. As a result, the sustainability of bilateral cooperation is weak as a whole.
Furthermore, to improve the service, the cross-border e-commerce enterprises need to cooperate with international logistics enterprises to provide high-quality services for consumers. This operation goal needs to be completed by both parties’ synchronous operation. But in the actual operation process, the cross-border e-commerce enterprises focus on their own work. To let consumers experience high-quality shopping services, they do not consider whether the work of international logistics enterprises is difficult or not. In the international logistics transportation services, the cross-border e-commerce enterprises will set the cooperation requirements of the nearest warehouse delivery, which limits the sustainable operation of international logistics enterprises. Considering the feasibility of international logistics delivery, human resources, transportation costs and other issues, international logistics enterprises are unable to meet the requirements of this international distribution, which results in contradictions in the operation and cooperation between both two parties.
In addition, there are still many problems worth being discussed between cross-border e-commerce trade and international logistics. Therefore, a larger number of scholars related in this field have had a attempt to study this proposition. Ying [1] performs a research about logistics mode selection in B2C cross-border e-commerce in China. His findings show that the lagged construction of international logistics system and the high cost of international logistics have seriously hindered the further development of China's B2C cross-border e-commerce enterprises. Even though with different samples like India, Alyoubi [2] finds that international logistics problems pose as a significant barrier for the sustainable development of e-commerce trade. However, Wang and Lee [3] sets China as a sample to discuss the effect of international logistics on international trade. They find that this kind of effect is positive. Meanwhile, Cho, et al [4] also find that the international logistics capability is positively correlated with the enterprises performance in terms of cross-border e-commerce market. Up to this day, although a great many of relating experts have deeply explored the dynamic relationship between cross-border e-commerce trade and international logistics, they have not reached a consensus. As a matter of fact, due to different perspectives, different samples, different times spans and different research approaches, drawing different conclusion about this proposition can be understood. Said differently, it is fully valuable to exploit the relationship between cross-border e-commerce trade and international logistics. because of this background, this paper sets Organization for Economic Co-operation and Development (OECD) countries as an example over the period 2000-2018 to explore the dynamic relationship between international logistics and cross-border e-commerce trade. Then, the panel data will be used to fulfil an empirical analysis under some econometric approaches such as panel unit root test, panel cointegration test, panel causality test and panel vector error correction model. Moreover, there are seven variables used in this paper. They are the cross-border e-commerce export trade, the cross-border e-commerce import trade, the GDP per capita, the population, the relative price, the international logistics and the one-period lagged outward foreign direct investment. Finally, all these variables and econometric methods will be employed to discuss the dynamic relationship between cross-border e-commerce trade and international logistics.
The contribution of this current paper to the previous empirical literature focusing on the relationship between cross-border e-commerce trade and international logistics is four fold. First, this paper attempts to fill the gap in the empirical literature with regard to the dynamic relationship between cross-border e-commerce trade and international logistics in the Organization for Economic Co-operation and Development (OECD) countries. The examination of the Organization for Economic Co-operation and Development (OECD) countries as a case study is particularly interesting since most of the Organization for Economic Co-operation and Development (OECD) countries are open enough to each other and fully integrated in economics, they have a high potential for performing economic activities from new trade modes and new logistics modes. Meanwhile, most of them have a huge capacity of producing a great quantity of goods and platforms to conduct the cross-border e-commerce trade. Moreover, most of them also have extremely advanced international logistics system to satisfy the needs of cross-border e-commerce trade activities. These basic fundamentals enable our results more reliable and robust. Second, unlike previous researches that have used the traditional econometric models such as vector auto-regression model and vector error correction model, this paper employs a recent multivariate economic technique which is famous as the panel data vector error correction model. Because the properties of both time series and cross-section of data are taken into consideration, this can help us more accurately to estimate the long-run and short-run relationship between cross-border e-commerce trade and international logistics. Third, a large number of panel unit root tests are used to test stationarity of variables used in this paper. This can also ensure the accuracy of empirical results. Finally, these evidences this paper provided can form a foundation for the future scholars who are interested in exploring the dynamic relationship between cross-border e-commerce trade and international logistics.
Moreover, I have used the sub-sector to proceed the Literature Review according your valuable advice.
- It is worth to refer to the works of the following authors:
Reply:
I have analyzed all these valuable researches you provided in the body of this paper.
- Need proper explanation of each table, such that reader can understand your work. Work especially on Tables 1-7.
Reply:
For <Table 1>:
<Table 1> presents the basic description of these variables used in this paper. The data includes 12544 observations for each variable. The mean of cross-border e-commerce export trade is 0.155 with a standard deviation equal to 0.096, minimum value equal to 0.052, maximum value equal to 0.464. The mean of cross-border e-commerce import trade is 0.098 with a standard deviation equal to 0.370, minimum value equal to 0.047, maximum value equal to 0.342. The mean of GDP per capita is 2.196 with a standard deviation equal to 3.172, minimum value equal to 4.745, maximum value equal to 14.053. The mean of population is 0.540 with a standard deviation equal to 0.702, minimum value equal to 0.336, maximum value equal to 1.235. The mean of relative price level is 2.397 with a standard deviation equal to 5.001, minimum value equal to -10.694, maximum value equal to 22.266. The mean of international logistics is 4.848 with a standard deviation equal to 0.749, minimum value equal to 1.416, maximum value equal to 5.145. The mean of one-lagged period outward foreign direct investment is 3.876 with a standard deviation equal to 0.940, minimum value equal to 0.326, maximum value equal to 4.512.
For <Table 2>:
<Table 2> presents these countries used in this paper. As we know, the Organization for Economic Co-operation and Development includes 38 countries. However, due to the data unavailability, only 30 countries are employed to testify the proposition of this paper.
For <Table 3>:
<Table 3> shows the results of five kinds of unit root tests. As for the results of IPS panel unit root test, the null hypotheses that the unit root does not exist are rejected. That is to say, all variables are non-stationary. However, taking the first difference, all of them turn stationary. As for the results of ADF panel unit root test, the null hypotheses that the unit root exists are rejected except the cross-border e-commerce export trade. After taking the first different, it becomes stationary. As for the results of PP panel unit root test, the null hypotheses that the unit root exists are rejected except the cross-border e-commerce export trade, cross-border e-commerce import trade, population and international logistics. Taking the first difference, all of them turn stationary. As for the LLC panel unit root test, the null hypotheses that the unit root exists are rejected except the cross-border e-commerce import trade. Taking the first difference, it turns stationary. As for the HADRI panel unit test, the null hypotheses that the unit root does not exist are rejected. This means that all variables are non-stationary. Taking the first difference, they turn stationary.
To summary, the estimating results indicate that most variables are not stationary at levels. However, when these variables are taken the first difference, all of them turn to be stationary. Therefore, it can be concluded that all these variables used in this paper are stationary after taking the first difference. So, it is essential to perform a panel cointegration test among cross-border e-commerce export trade, cross-border e-commerce import trade, GDP per capita, population, relative price level, international logistics and one-period lagged outward foreign direct investment.
For <Table 4>:
<Table 4> presents the results of cointegration test for model (1). Both Trace test and Maximum Eigen-value test verify that the null hypothesis that no cointegration exists is rejected. Said differently, the cointegration exists for model (1). Meanwhile, the results also indicate the international logistics is positive related to cross-border e-commerce export trade and significant in statistic. Specifically speaking, 1% increase in international logistics leads to 0.025% increase in cross-border e-commerce export trade. Moreover, this result verifies Hypothesis One. What’s more, this result is also consistent with findings of [35, 36].
For <Table 5>:
<Table 5> presents the results of cointegration test for model (2). Both Trace test and Maximum Eigen-value test verify that the null hypothesis that no cointegration exists is rejected. Said differently, the cointegration exists for model (2). At the same time, the results also indicate the cross-border e-commerce export trade is positively related to international logistics and significant in statistic. Concretely speaking, 1% increase in cross-border e-commerce export trade results in 2.681% increase in international logistics. Furthermore, this result also verifies Hypothesis One. Additionally, this result is in keeping with results of [3, 37].
For <Table 6>:
<Table 6> presents the results of cointegration test for model (3). Both Trace test and Maximum Eigen-value test verify that the null hypothesis that no cointegration exists is rejected. Said differently, the cointegration exists for model (3). In the meanwhile, the results also indicate the international logistics has a positive effect on cross-border e-commerce import trade and significant in statistic. Concretely speaking, 1% increase in international logistics brings about 0.017% increase in cross-border e-commerce import trade. In addition, this result also verifies Hypothesis Three. Moreover, this result is in accordance with ideas of [38, 39].
For <Table 7>:
<Table 7> presents the results of cointegration test for model (4). Both Trace test and Maximum Eigen-value test verify that the null hypothesis that no cointegration exists is rejected. Said differently, the cointegration exists for model (4). Meanwhile, the results also indicate the cross-border e-commerce import trade has a positive effect on international logistics and significant in statistic. Say concretely, 1% increase in cross-border e-commerce import trade generates 1.036% increase in international logistics. In addition, this result also verifies Hypothesis Three. Moreover, this result is identical to the findings of [42-40].
- In the Discussion section, the obtained results should be confronted with the opinions contained in the literature on the subject and with the results of the other studies, so that it is clear whether the obtained results confirm or question the current state of knowledge.
Reply:
I have modified this part according your valuable advice in the body of this paper.
- The basic pillars of the sustainability were not maintained by your research. Correct it.
Reply:
I have corrected this part according your valuable advice for the full paper.
- Summarizing the article should be refined in the following areas: • Expand the theoretical part and enrich the research part with discussion, • Develop conclusions; The description of the methodology used and the different steps taken may be improved.
Reply:
I have modified this part according your valuable advice for the full paper one by one. Specific details are reflected in the manuscript.

Reviewer 4 Report
Very interesting article. It contains many innovative elements. Even so, I have some comments as the article is a bit messy in some places:
- In the abstract, please clearly state the purpose of the work. Please note that the abstract can be up to 200 words long.
- Why do the conclusions appear in the introduction? Conclusions should be at the end of the work.
- The introduction also lacks an unambiguously formulated aim of the work.
- Line 128: "The dry sports" ??? it's probably a typo.
- A very modest literature review. It would be good to widen it. To do this, for example, international logistics and cross-border e-commerce can be defined.
- Why are the web pages of lines 205-208 not in the list of references? Please complete the.
- Patterns 1-4: what is this model? You write what it contains and what it describes, but you don't name the model. Did you make it up yourself? If not, there must be an introduction. It is also good to give the source of such a model. It will also allow the literature to expand. Please check if such descriptions are available for each model.
- Lines 201-208 should be in the introduction and not in the model description. It is from the introduction that the reader should find out where the data comes from.
- "This paper sets OECD countries as an example to explore the influential mechanism between international logistics and cross-border e-commerce trade including export and import. the panel data from 2000 to 2018 will be employed to conduct an empirical analysis under a series of econometric approaches such as panel unit root test and panel vector error correction model ”(lines 322-325). It was already. There is no reason to keep writing this. Write once in the introduction and that's enough. Please check the full article carefully.
- Please check all patterns carefully to see if there is any mistake.
Author Response
- In the abstract, please clearly state the purpose of the work.
Unlike previous papers in the international logistics - cross-border e-commerce trade, this paper sets Organization for Economic Co-operation and Development countries as an example to explore the dynamic interaction between international logistics and cross-border e-commerce trade.
- Please note that the abstract can be up to 200 words long.
Unlike previous papers in the international logistics - cross-border e-commerce trade, this paper sets Organization for Economic Co-operation and Development countries as an example to explore the dynamic interaction between international logistics and cross-border e-commerce trade. The panel data over the period 2000-2018 will be employed to fulfil an empirical analysis under a host of econometric techniques such as panel unit root test, panel cointegration test, panel causality test and panel vector error correction model. Incorporating with other control variables, we find that there is a long-run relationship between international logistics and cross-border e-commerce trade. Specifically speaking, in the long run, the international logistics has a positive and significant effect on cross-border e-commerce trade. However, in the short run, the international logistics has a negative and significant effect on cross-border e-commerce trade. Furthermore, the results suggest that the deviation from cointegration system of cross-border e-commerce trade and international logistics will lead to the cross-border e-commerce trade and international logistics to change with a range of approximately 2.2% to 47.2% in the next period. Therefore, referring to these findings, each OECD countries’ government should take corresponding policies to make a sustainable development of both international logistics and cross-border e-commerce trade.
- Why do the conclusions appear in the introduction? Conclusions should be at the end of the work.
I have fully deleted this part.
- The introduction also lacks an unambiguously formulated aim of the work.
In reality, the sustainable development of both international logistics and cross-border e-commerce trade also faces many challenges. From the theoretical level, the rapid development of cross-border e-commerce trade can increase the transaction orders, which can promote the sustainable development of international logistics. However, in the actual cooperation, the coordination and sustainability of both international logistics and cross-border e-commerce trade are fragile. As we know, the e-commerce platform uses the Internet to establish the communication links with consumers, which provides consumers with more convenient product purchase channels. Transactions can be achieved only with the help of the network, which is not limited by both space and time. As a platform transportation partner, the international logistics needs to do a good job in customs such as quality inspection and other work after receiving the international logistics transportation requirements sent by the platform. As a matter of fact, each link is very complex and takes a lot of time. To solve the time-consuming problem, some cross-border e-commerce trade service providers choose to establish overseas storage model to offer customers with return and exchange services. Due to the large cost of overseas warehousing construction, the demand for product sales data information is high. At present, the international logistics enterprises can not obtain the product warehouse management data information, which leads to the low accuracy of warehouse design. The layout of storage space can not meet the actual storage needs, which is not conducive to the improvement of warehouse management level. And the ability of international logistics enterprises has always been difficult to maintain the sustainable development. However, the cross-border e-commerce platform does not know enough about overseas warehousing, and does not understand the importance of international logistics enterprises to their own sustainable development. As a result, the sustainability of bilateral cooperation is weak as a whole.
Furthermore, to improve the service, the cross-border e-commerce enterprises need to cooperate with international logistics enterprises to provide high-quality services for consumers. This operation goal needs to be completed by both parties’ synchronous operation. But in the actual operation process, the cross-border e-commerce enterprises focus on their own work. To let consumers experience high-quality shopping services, they do not consider whether the work of international logistics enterprises is difficult or not. In the international logistics transportation services, the cross-border e-commerce enterprises will set the cooperation requirements of the nearest warehouse delivery, which limits the sustainable operation of international logistics enterprises. Considering the feasibility of international logistics delivery, human resources, transportation costs and other issues, international logistics enterprises are unable to meet the requirements of this international distribution, which results in contradictions in the operation and cooperation between both two parties.
In addition, there are still many problems worth being discussed between cross-border e-commerce trade and international logistics. Therefore, a larger number of scholars related in this field have had a attempt to study this proposition. Ying [1] performs a research about logistics mode selection in B2C cross-border e-commerce in China. His findings show that the lagged construction of international logistics system and the high cost of international logistics have seriously hindered the further development of China's B2C cross-border e-commerce enterprises. Even though with different samples like India, Alyoubi [2] finds that international logistics problems pose as a significant barrier for the sustainable development of e-commerce trade. However, Wang and Lee [3] sets China as a sample to discuss the effect of international logistics on international trade. They find that this kind of effect is positive. Meanwhile, Cho, et al [4] also find that the international logistics capability is positively correlated with the enterprises performance in terms of cross-border e-commerce market. Up to this day, although a great many of relating experts have deeply explored the dynamic relationship between cross-border e-commerce trade and international logistics, they have not reached a consensus. As a matter of fact, due to different perspectives, different samples, different times spans and different research approaches, drawing different conclusion about this proposition can be understood. Said differently, it is fully valuable to exploit the relationship between cross-border e-commerce trade and international logistics. because of this background, this paper sets Organization for Economic Co-operation and Development (OECD) countries as an example over the period 2000-2018 to explore the dynamic relationship between international logistics and cross-border e-commerce trade.
- Line 128: "The dry sports" ??? it's probably a typo.
Reply:
I have corrected this mistake in the body of this paper.
- A very modest literature review. It would be good to widen it. To do this, for example, international logistics and cross-border e-commerce can be defined.
Reply:
I have enriched the literature Review. Derails are presented in the body of this paper.
- Why are the web pages of lines 205-208 not in the list of references? Please complete the.
Reply:
I have completed this part in the body of this paper.
- Patterns 1-4: what is this model? You write what it contains and what it describes, but you don't name the model. Did you make it up yourself? If not, there must be an introduction. It is also good to give the source of such a model. It will also allow the literature to expand. Please check if such descriptions are available for each model.
Reply:
Model (1) - model (4) are established by the authors of this paper. I have given description of each model.
- Lines 201-208 should be in the introduction and not in the model description. It is from the introduction that the reader should find out where the data comes from.
Reply:
I have revised this part according to your valuable advice. Meanwhile, all the data collection websites have been provided in the body of this paper.
- "This paper sets OECD countries as an example to explore the influential mechanism between international logistics and cross-border e-commerce trade including export and import. the panel data from 2000 to 2018 will be employed to conduct an empirical analysis under a series of econometric approaches such as panel unit root test and panel vector error correction model ”(lines 322-325). It was already. There is no reason to keep writing this. Write once in the introduction and that's enough. Please check the full article carefully.
Reply:
I have revised this part according to your valuable advice. Derails are presented in the body o this paper.
- Please check all patterns carefully to see if there is any mistake.
Reply:
I have checked and corrected all mistakes .

Round 2
Reviewer 1 Report
All suggestions have been taken into account.
Reviewer 3 Report
Thanks for the corrections and clarifications.
Congratulations.
Reviewer 4 Report
Thank you for following the directions. Now the article is much better.